# DISTRIBUTIONAL MACHINE UNLEARNING
# VIA SELECTIVE DATA REMOVAL

**Youssef Allouah**[1*]**, Rachid Guerraoui**[2]**, Sanmi Koyejo**[1]
[1]Stanford University, USA  [2] EPFL, Switzerland
{yallouah,sanmi}@cs.stanford.edu

## ABSTRACT

Machine learning systems increasingly face requirements to remove entire domains of information—such as toxic language or biases—rather than individual user data. This task presents a dilemma: full removal of the unwanted domain data is computationally expensive, while random partial removal is statistically inefficient. We find that a domain's statistical influence is often concentrated in a small subset of its data samples, suggesting a path between ineffective partial removal and unnecessary complete removal. We formalize this as *distributional unlearning*: a framework to select a small subset that balances forgetting an unwanted distribution while preserving a desired one. Using Kullback-Leibler divergence constraints, we derive the exact removal-preservation Pareto frontier for Gaussian distributions and prove that models trained on the edited data achieve corresponding log-loss bounds. We propose a distance-based selection algorithm and show it is quadratically more sample-efficient than random removal in the challenging low-divergence regime. Experiments across synthetic, text, and image datasets (Jigsaw, CIFAR-10, SMS spam) show our method requires 15–82% less deletion than full removal for strong unlearning effects, e.g., halving initial forget set accuracy. Ultimately, by showing a small forget set often suffices, our framework lays the foundations for more scalable and rigorous subpopulation unlearning.

## 1 INTRODUCTION

As machine learning models persist in production, the data on which they were trained often becomes legally or ethically objectionable, requiring deletion. The nature of these requests is now scaling beyond individual user data towards erasing entire subpopulations, which can represent unwanted domains, concepts, or any data-defined construct, e.g., harmful biases or toxic language (Kurmanji et al., 2024). This new scale of unlearning presents a dilemma. On one hand, removing the full set of domain data involves a prohibitive cost, since the computational cost of efficient removal methods typically scales with the size of the forget set (Guo et al., 2020). This challenge is highlighted in efforts to unlearn creative works like the Harry Potter book series (Eldan & Russinovich, 2023), where the sheer size of the identified text and the trained model make the subsequent unlearning action computationally expensive. On the other hand, removing a random small subset can be statistically inefficient because the statistical footprint of the domain persists. For instance, large language models can still verbatim reproduce training sequences even if the specific source text is removed (Liu et al., 2025), due to overlapping context with the remaining data. This dilemma, between the expensive cost of full removal and the ineffectiveness of naive partial removal, leaves a methodological gap for a more targeted and efficient approach to domain unlearning.

To address this methodological gap, we start with a key observation: a domain's statistical influence is often concentrated in a small, high-impact subset of its samples. This suggests an optimal strategy exists between the two extremes: a targeted removal of only this impactful subset, which can be both computationally efficient and statistically effective. To formalize this, we model domains as unknown probability distributions, a well-precedented abstraction in both statistical learning (Shalev-Shwartz & Ben-David, 2014) and natural language processing (Blei et al., 2003; Srivastava & Sutton, 2017). This statistical framing allows us to pose a precise research question: *what is the minimal set of data points to remove to make the data distribution far from an unwanted domain, yet simultaneously close to a retained one?* We introduce *distributional unlearning*, an information-theoretic framework that quantitatively addresses this question. This data-centric,

---

*Work done while at EPFL. Code available at: https://github.com/ysfalh/unlearning-distribution.

statistical framing is complementary to the rich literature on what we term *sample-level* unlearning: methods that efficiently approximate retraining without a pre-defined set of individual records. While prior work on sample-level methods, based on influence functions (Guo et al., 2020) or data sharding (Bourtoule et al., 2021), have made important progress on the *computational* problem, they do not address the *statistical* question of which samples should constitute that set for maximal impact. Similarly, while class-level unlearning (Tarun et al., 2023; Kodge et al., 2024) and concept erasure (Ravfogel et al., 2020; Belrose et al., 2023) methods also operate on certain domains, they either act on a model's internal representations—making the changes potentially reversible—or lack a formal method for finding a small subset to remove. This leaves our foundational question unanswered: which samples to remove in the first place?

**Contributions.** We summarize our contributions in tackling the above question as follows:

- *Framework:* We formalize distributional unlearning, a data-centric framework that uses Kullback-Leibler divergence constraints to balance removal and preservation guarantees.
- *Pareto frontier & downstream guarantees:* We derive the closed-form Pareto frontier of achievable removal-preservation levels for exponential families. We further prove our framework provides log-loss shift guarantees for downstream models.
- *Algorithm & sample efficiency:* We study the sample complexity of distributional unlearning with both random and selective removal mechanisms. We propose a distance-based selection algorithm and prove that it achieves a quadratic improvement in sample efficiency over random removal in low-divergence Gaussian regimes.
- *Empirical validation:* We validate our framework on synthetic, text, and image data (Jigsaw, CIFAR-10, SMS spam). While our synthetic analysis suggests high potential data efficiency gains up to 82%, real-world data complexity reduces these to a still-significant 15–50% in deletion budget savings. We show these data savings to hold in combination with various downstream sample-level unlearning methods.

## 1.1 RELATED WORK

**Sample-level unlearning.** Sample-level unlearning has made impressive strides in fast model updates and formal deletion guarantees (Neel et al., 2021; Zhang et al., 2024a; Chien et al., 2024; Allouah et al., 2025; Koloskova et al., 2025). Izzo et al. (2021) use influence-function updates to approximate the effect of deleting a single point without full retraining; Guo et al. (2020) provide certified bounds on how close the post-deletion model is to a scratch-retrained one; Bourtoule et al. (2021) employ data sharding to efficiently erase small batches. However, this class of methods does not address which or how many samples must be removed to eliminate a domain's overall statistical footprint. Our work complements these techniques by asking not just how to update a model once samples are flagged, but which samples to flag in the first place, and in what quantity.

**Concept erasure.** Concept erasure methods tackle unwanted attributes in learned features. INLP (Ravfogel et al., 2020) repeatedly projects out directions predictive of a protected attribute; counterfactual augmentation (Kaushik et al., 2020) synthesizes targeted data to sever causal links; adversarial training (Elazar & Goldberg, 2018) trains encoders to remove specific signals. These operate post-hoc on a fixed model's representations—ideal for fairness use-cases such as removing gender—but they rely on white-box access and tailor to one model at a time. Where concept erasure edits model representations, we edit the data, guaranteeing forgetting for downstream models.

**Coresets & domain adaptation.** Domain adaptation theory (Ben-David et al., 2010) seeks to minimize the impact of divergence between domains; we flip this paradigm by intentionally increasing divergence from an unwanted source while controlling proximity to a desired one. Similarly, our work inverts the goal of coresets, which approximate a single distribution for efficient training (Mirzasoleiman et al., 2020; Gentile et al., 2024). Coreset methods are designed for a one-distribution problem, whereas we select samples from an unwanted distribution based on their relationship to a separate, retained distribution. This two-distribution approach is critical; our experiments show that a coreset-based baseline is inefficient for unlearning because it ignores the retain data.

## 2 DISTRIBUTIONAL UNLEARNING: DEFINITION AND IMPLICATIONS

The predominant paradigm in machine unlearning, which we call sample-level unlearning, addresses the computational challenge of fully removing a known finite set of data points. This work addresses

a *complementary* scenario, which we call distributional unlearning, where the goal is to erase the statistical influence of a subpopulation. In this section, we formalize this task, establish its fundamental trade-offs, and prove its consequences for downstream predictors.

**Problem Statement.** To make the abstract concept of the unwanted domain tractable, we first model it as an underlying probability distribution $p_1$. This abstraction, common in statistical machine learning, allows us to set a mathematical objective: to construct a new data distribution $p$ that is statistically distant from $p_1$ yet remains close to a retained distribution $p_2$. In practice, these true distributions are unknown. We instead work with finite sets of samples: $S_1 = \{x_i^{(1)}\}_{i=1}^{n_1}$ drawn i.i.d. from unwanted distribution $p_1$, and $S_2 = \{x_j^{(2)}\}_{j=1}^{n_2}$ from retained distribution $p_2$. Crucially, we assume these sets are obtained via some upstream process, such as keyword filtering (as in our Jigsaw experiments), the output of a pre-trained classifier, or human annotation. Our contribution is to solve the subsequent statistical problem: given these identified samples, which subset should be removed to most efficiently achieve our objective defined at the distribution level?

To formalize our objective, we choose Kullback-Leibler (KL) divergence because it enables control over the expected log-loss of downstream models, hence connecting data-level edits to predictive outcomes. We recall for two absolutely-continuous distributions $q, p$ on data space $\mathcal{X}$ that $\mathrm{KL}(q\|p) := \int_{\mathcal{X}} q(x) \log \frac{q(x)}{p(x)} \, dx$.

**Definition 1** (($\alpha, \varepsilon$)-Distributional Unlearning). *For tolerances $\alpha, \varepsilon > 0$, a distribution $p \in \mathcal{P}$ satisfies ($\alpha, \varepsilon$)-distributional unlearning with respect to $(p_1, p_2)$ if:*

$$\mathrm{KL}(p_1 \| p) \geq \alpha \quad \text{(removal)}, \qquad \mathrm{KL}(p_2 \| p) \leq \varepsilon \quad \text{(preservation)}. \qquad (1)$$

The first inequality forces the edited data to be information-theoretically distant from the population we wish to forget. The second inequality upper bounds collateral damage to the population we preserve. While we focus on KL divergence for its analytical tractability and its direct control of expected log-loss, we note that different tasks may favor different divergences, and our framework can in principle accommodate alternative or task-weighted notions of discrepancy. In the remainder of this section, we analyze the properties of this definition at the population level, capturing the fundamental trade-offs and downstream learning-theoretic guarantees. In Section 3, we turn to the practical finite-sample setting and provide high-probability guarantees for specific removal algorithms. Throughout, we defer all proofs to Appendix A.

**Removal-Preservation Trade-off.** The pair $(\alpha, \varepsilon)$ captures a trade-off: how far we can move from $p_1$ while remaining close to $p_2$. To understand which $(\alpha, \varepsilon)$ pairs are jointly achievable, we characterize the feasible region and its boundary. Formally, the *Pareto frontier* $\mathrm{PF}(p_1, p_2; \mathcal{P})$ consists of those pairs $(\alpha, \varepsilon)$ for which no strictly better trade-off exists: there is no $p' \in \mathcal{P}$ satisfying $\mathrm{KL}(p_1\|p') \geq \alpha'$ and $\mathrm{KL}(p_2\|p') \leq \varepsilon'$ with $\alpha' > \alpha$ and $\varepsilon' < \varepsilon$. That is, every point on the frontier is optimal in the sense that one objective cannot be improved without worsening the other. To build intuition, we first derive this frontier for the analytically tractable case of Gaussian distributions.

**Proposition 1** (Pareto Frontier). *Let $p_1, p_2$ be two distributions in $\mathcal{P}$, the class of Gaussian distributions with shared positive covariance. The Pareto frontier of $(\alpha, \varepsilon)$ values achievable in $\mathcal{P}$ is:*

$$\mathrm{PF}(p_1, p_2; \mathcal{P}) = \left\{ \left( \alpha, \left( \sqrt{\alpha} - \sqrt{\mathrm{KL}(p_1\|p_2)} \right)^2 \right) : \alpha \geq \mathrm{KL}(p_1\|p_2) \right\}.$$

This frontier quantifies a natural, fundamental cost of distributional unlearning: a minimal preservation loss is incurred for any given removal level, a trade-off governed by the initial divergence $\mathrm{KL}(p_1\|p_2)$. The result also highlights the suboptimality of a common default: keeping only the retained data $p_2$. While this strategy achieves perfect preservation, the frontier shows it is possible to attain a significantly higher level of removal by accepting a potentially small preservation loss. While shown here for Gaussians for simplicity, this trade-off holds more generally for regular exponential families (Appendix A, Theorem 4) and is validated by our synthetic experiments (Fig. 1).

**Downstream Performance.** We now connect our data-level objective to predictive performance in supervised learning. Consider a predictor $h : \mathcal{X} \to \Delta(\mathcal{Y})$, where $\mathcal{X}$ is the input space and $\Delta(\mathcal{Y})$ the probability simplex over label space $\mathcal{Y}$, trained on a distribution $p$ over $\mathcal{X} \times \mathcal{Y}$ that satisfies ($\alpha, \varepsilon$)-distributional unlearning with respect to $(p_1, p_2)$. We study how $h$ performs under the true data-generating distributions $p_1$ and $p_2$.
Let $\ell(y, q) := -\log q(y)$, for $y \in \mathcal{Y}, q \in \Delta(\mathcal{Y})$, denote the log-loss. Define the expected loss under $p$ as $\mathcal{L}(h; p) := \mathbb{E}_{(x,y) \sim p}[\ell(y, h(x))]$. Then, for any class of distributions $\mathcal{P}$, we have:

| Removal Method | Sample Complexity | |
| --- | --- | --- |
| | Removal | Preservation |
| Random (Prop. 3) | $n_1\left(1 - \sqrt{1-\alpha}\right)$ | $n_1\left(1 - \sqrt{\varepsilon}\right)$ |
| Selective (Thm. 1) | $n_1\left(1 - (1-\alpha)^{1/4}\right)$ | $n_1\left(1 - \varepsilon^{1/4}\right)$ |

Table 1: Summary of *simplified* sample complexity bounds, showing the number of $p_1$ samples (out of $n_1 \geq 1$) to remove to achieve $(\alpha, \varepsilon)$-distributional unlearning with high probability. We assume $n_2$ is large, $\text{KL}(p_1\|p_2)$ is small, and $\frac{n_2}{n_1}$ is constant; see Corollary 10 for details. The key insight is the quadratic improvement is sample efficiency of selective removal over random in this regime.

**Proposition 2.** *Let $h$ minimize $\mathcal{L}(h; p)$, and let $h_1$, $h_2$ be optimal predictors under $p_1, p_2 \in \mathcal{P}$, respectively. If $p$ satisfies $(\alpha, \varepsilon)$-distributional unlearning with respect to $(p_1, p_2)$, then:*

$$\mathcal{L}(h; p_1) - \mathcal{L}(h_1; p_1) \geq \alpha - \delta_1, \qquad \mathcal{L}(h; p_2) - \mathcal{L}(h_2; p_2) \leq \varepsilon - \delta_2, \qquad (2)$$

*where $\delta_1 := \text{KL}(p_{1,X}\|p_X), \delta_2 := \text{KL}(p_{2,X}\|p_X)$ denote the marginal KL divergence over inputs.*

These bounds show that distributional unlearning guarantees increased loss under the forgotten distribution and bounded degradation under the preserved one. In this sense, our framework provides meaningful control over downstream predictive behavior. Regarding the extra marginal KL term in the first inequality, which quantifies divergence on input distributions, the data-processing inequality gives $\text{KL}(p_{1,X}\|p_X) \leq \text{KL}(p_1\|p)$, hence this extra term is always bounded by the same $\alpha$ we already control. A similar term appears in the second inequality, but can only improve the preservation bound. Finally, these bounds provide a practical rule for calibrating the $(\alpha, \varepsilon)$ parameters. For instance, if a practitioner decides they can tolerate at most a $0.1$ nat increase in log-loss on the retained test set, they can set their preservation budget to $\varepsilon = 0.1$. Then, the Pareto frontier (Prop. 1) indicates the maximum achievable removal for this budget, in the Gaussian case. In a scenario where the initial distributions have divergence $\text{KL}(p_1\|p_2) = 2$, this choice of $\varepsilon$ corresponds to the removal target of $\alpha = 3$. These values match our findings in controlled experiments (Fig. 2, 2nd column).

# 3 ALGORITHMS AND SAMPLE COMPLEXITY

The previous section defined our framework at the population level. We now turn to the practical finite-sample setting, where we must achieve $(\alpha, \varepsilon)$-distributional unlearning using only the drawn data samples introduced in Section 2. To build an analytical understanding of the finite-sample behavior, we analyze the problem in an idealized but foundational setting: univariate Gaussian distributions with known variance. This tractable setting allows us to derive closed-form sample complexity bounds, providing crucial intuition about the relative efficiency of different removal strategies. We then validate that these insights generalize empirically to more complex settings in Section 4.

In the following, we introduce and analyze two deletion strategies: a random baseline and a selective, distance-based method. We focus on the class of distributions $\mathcal{P} := \{\mathcal{N}(\mu, \sigma^2): \mu \in \mathbb{R}\}$, with known $\sigma > 0$. Given $n_1$ i.i.d. samples from the unwanted distribution $p_1 \in \mathcal{P}$ and $n_2$ from the retained distribution $p_2 \in \mathcal{P}$, and a deletion budget $0 \leq f \leq n_1$, we derive high-probability bounds on the resulting $(\alpha, \varepsilon)$-distributional unlearning guarantees. We defer all proofs to Appendix A.

## 3.1 RANDOM REMOVAL

We begin with a baseline deletion strategy that treats every sample equally, deleting $f$ points chosen uniformly at random from the $n_1$ samples of $p_1$. The formal procedure is as follows:

**Algorithm (Random Removal).**

1. Randomly select $f$ out of the $n_1$ samples of $p_1$ without replacement.
2. Remove those $f$ samples.
3. Re-fit $\mathcal{N}(\hat{\mu}, \sigma^2)$ by MLE (maximum likelihood estimation) on the remaining data.

The following theorem provides a finite-sample guarantee for achieving $(\alpha, \varepsilon)$-distributional unlearning using random removal with a deletion budget $f$.

**Proposition 3** (Random Removal). *Let $p_1, p_2 \in \mathcal{P}$ and $\delta \in (0, 1)$. We observe $n_1$ samples from $p_1$ and $n_2$ samples from $p_2$, and randomly remove $f$ samples from $p_1$ before fitting. With probability*

$1 - \delta$, *the resulting MLE distribution satisfies* $(\alpha, \varepsilon)$-*distributional unlearning with:*

$$\alpha \geq \left(\frac{1}{2} - 3\left(\frac{n_1 - f}{n_2}\right)^2\right) \mathrm{KL}(p_1 \parallel p_2) - \frac{3\ln(4/\delta)}{2n_2}\left(1 + \frac{n_1 - f}{n_2}\right),$$

$$\varepsilon \leq 3\left(\frac{n_1 - f}{n_2}\right)^2 \mathrm{KL}(p_1 \parallel p_2) + \frac{3\ln(4/\delta)}{n_2}\left(1 + \frac{n_1 - f}{n_2}\right).$$

This result shows that the effectiveness of random removal is driven by the ratio of remaining unwanted samples to retained samples $\frac{n_1 - f}{n_2}$. An interesting aspect of these bounds is the quadratic dependence on this ratio, which indicates diminishing returns: each subsequent random deletion provides progressively less of an unlearning effect. This quadratic relationship stems from the concentration of the empirical mean, whose variance scales inversely with the sample size. While conceptually simple, this method's inefficiency arises because it treats all samples equally, failing to prioritize those that contribute most to the unwanted statistical patterns.

### 3.2 SELECTIVE REMOVAL

We hypothesize that a more effective strategy than random removal should prioritize which samples to delete, using $p_2$ as reference. Since our goal is to shift the dataset's empirical mean away from the unwanted center $\mu_1$ and towards the retained center $\mu_2$, the most impactful samples to remove are those from $p_1$ that are furthest from $\mu_2$. This intuition leads to our proposed selective removal strategy, which uses the empirical mean of the retained data $\hat{\mu}_2$ as a reference point for selection.

**Algorithm (Selective Removal).**

1. Compute the mean $\hat{\mu}_2$ of the $n_2$ samples from $p_2$.
2. For each of the $n_1$ samples $x_i$ from $p_1$, compute the score $s_i = |x_i - \hat{\mu}_2|$.
3. Delete the $f$ samples with the largest scores $s_i$.
4. Re-fit $\mathcal{N}(\hat{\mu}, \sigma^2)$ by MLE on the remaining data.

The following theorem provides a finite-sample guarantee for achieving $(\alpha, \varepsilon)$-distributional unlearning using selective removal with a deletion budget $f$.

**Theorem 1** (Selective Removal). *Let* $p_1, p_2 \in \mathcal{P}$ *and* $\delta \in (0, 1)$. *Let* $f$ *samples from* $p_1$ *be removed according to Selective Removal. With probability* $1 - \delta$, *the resulting estimate satisfies* $(\alpha, \varepsilon)$-*distributional unlearning with:*

$$\alpha \geq \frac{1}{2}\mathrm{KL}(p_1 \parallel p_2) - \frac{1}{2}\left(\frac{n_1 - f}{n_2}\right)^2 g^{-1}\left(1 - \frac{f}{n_1} + \sqrt{\frac{\ln(4/\delta)}{2n_1}}; \mathrm{KL}(p_1 \parallel p_2)\right)^2 - \frac{\ln(4/\delta)}{n_2},$$

$$\varepsilon \leq \left(\frac{n_1 - f}{n_2}\right)^2 g^{-1}\left(1 - \frac{f}{n_1} + \sqrt{\frac{\ln(4/\delta)}{2n_1}}; \mathrm{KL}(p_1 \parallel p_2)\right)^2 + \frac{2\ln(4/\delta)}{n_2},$$

*where* $g(u; \kappa) := \Phi(u - \sqrt{2\kappa}) + \Phi(u + \sqrt{2\kappa}) - 1$, *for* $u, \kappa > 0$, *and* $\Phi$ *is the standard normal CDF.*

While the above expression is more complex than that of Prop. 3, it reveals a significant improvement in efficiency, materialized in the term involving the inverse CDF $g^{-1}(\cdot; \kappa)$. Intuitively, this term represents a quantile of a folded normal distribution, shifted by $\kappa = \mathrm{KL}(p_1 \parallel p_2)$, and arises because we are truncating the distribution of scores by removing those in the tail. Specifically, this term strictly amplifies the quadratic decrease in $f$, which was the best we could previously obtain with random removal. This amplification is greatest when the distributions are close (the low-divergence regime), as the "outlier" samples are more distinct and their removal provides a greater and more targeted shift in the empirical mean. As we summarize in Table 1 and derive formally in Corollary 10 (Appendix A), this improved leverage translates directly into a quadratic improvement in sample efficiency over the random baseline in low-divergence regimes. This theoretical advantage is a core finding of our work and is empirically validated in our experiments (Fig. 2).

## 4 EMPIRICAL VALIDATION

We now empirically evaluate our distributional unlearning framework across several case studies, moving from synthetic to more complex real-world data. These experiments are designed to validate

| Domain | Separability | Target on $p_1$ | Random | Selective | Savings |
|---|---|---|---|---|---|
| Gaussians | Low | $KL(p_1\|p)$ | 65 | 18 | **82 %** |
| Gaussians | High | $KL(p_1\|p)$ | 65 | 50 | **50 %** |
| Jigsaw toxic comments | Low | Recall | 100 | 85 | **15 %** |
| SMS Spam | Medium | Recall | 90 | 75 | **25 %** |
| CIFAR-10 | High | Accuracy | 80 | 50 | **50 %** |

Table 2: Deletion budget (%) needed to reach half of the initial value of the removal metric (no deletion) on each dataset. "Selective"=best-performing selective removal score; "Saving"=relative size reduction versus full removal, i.e. retrain on $p_2$ samples only. Gaussians (low) and (high) are the scenarios of the top leftmost and rightmost plots of Fig. 2, respectively. "Separability"=summarizes how distinguishable the domains are; the observed savings follow this difficulty.

the qualitative trends predicted by our theory—notably, the superior sample efficiency of selective removal—rather than to directly apply Theorem 1 and Proposition 3, which assume Gaussianity. To this end, we operationalize abstract domains using well-defined proxies, e.g., keyword-based subsets, image classes. This allows for a reproducible evaluation of our selection algorithms, while acknowledging that the upstream task of identifying such domains in the wild is a separate challenge.

Our validation begins with synthetic Gaussians to directly verify our theoretical predictions in a controlled environment. We then move to high-dimensional real-world data, starting with the case of well-separated distributions (CIFAR-10), showing our framework generalizes standard class unlearning. We then test our framework in the more challenging scenario of intertwined distributions (Jigsaw toxic comments), where the unwanted domain is semantically linked to the main task. Finally, we demonstrate the broad applicability of our data-centric approach by using it as an efficiency-boosting front-end for existing sample-level unlearning algorithms. We defer results on an additional text dataset (SMS Spam) and full experimental details to Appendix B.

**Results overview.** We summarize our main empirical findings in Table 2. Our synthetic experiments directly validate our theory, showing data savings of up to 82% in the low-divergence Gaussian regime, where our analysis predicts the greatest advantage. Crucially, this insight generalizes to high-dimensional non-Gaussian text and image data, where selective methods still provide significant 15–50% data savings. As expected, the magnitude of these gains is more modest than in the idealized theoretical setting, reflecting the increased complexity of real-world distributions. Across all experiments, these removal gains are achieved with negligible impact on performance on the retained domains, confirming the downstream performance guarantees predicted by Proposition 2. In particular, the intertwined distribution case study on Jigsaw confirms that simply deleting all $p_1$ is suboptimal, as discussed after Proposition 1. Finally, we show in Table 5 that our selection methods combine effectively with various sample-level unlearning methods, not just retraining from scratch.

### 4.1 SYNTHETIC GAUSSIANS: PARETO FRONTIER AND SAMPLE EFFICIENCY

We first validate our theoretical framework in a controlled setting, designed to directly verify the analytical predictions made in Sections 2 and 3: the Pareto frontier shape and the superior sample efficiency of selective removal. We set $p_1 = \mathcal{N}(0, 1)$ and $p_2 = \mathcal{N}(\mu, 1)$ for varying $\mu \in \{0.5, 2.5, 5\}$, which allows controlling the initial divergence $KL(p_1\|p_2)$, i.e., intertwinement level. For each configuration, we draw $n_1 = n_2 = 1000$ samples from $p_1$ and $p_2$, respectively. We implement the two mechanisms from Section 3: *random* and *selective* removal. After deleting $f$ samples, we re-fit a Gaussian $p = \mathcal{N}(\hat{\mu}, 1)$ on the retained $p_1$ and $p_2$ samples. We then compute forward KL divergences $\alpha = KL(p_1\|p)$ and $\varepsilon = KL(p_2\|p)$ to quantify forgetting and preservation, respectively.

**Pareto frontier.** Figure 1 confirms that the empirical $(\alpha, \varepsilon)$ trade-off closely matches the theoretical Pareto frontier derived in Proposition 1. To plot feasible empirical trade-offs, we set $p = \mathcal{N}(\mu, 1)$ and vary $\mu \in \mathbb{R}$, while $p_1 = \mathcal{N}(0, 1)$ and $p_2 = \mathcal{N}(2, 1)$ so that $KL(p_1\|p_2) = 2$. The latter quantity is the threshold predicted by the theory, and validated by Figure 1. Indeed, feasible trade-offs whose removal divergence $\alpha$ is below this threshold are pareto-suboptimal. They are dominated by the trade-off $(\alpha = KL(p_1\|p_2), \varepsilon = 0)$, which can be achieved with the choice of distribution $p = p_2$.

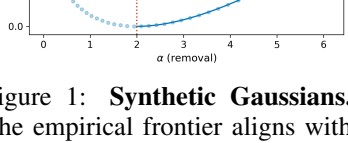

Figure 1: **Synthetic Gaussians.** The empirical frontier aligns with the theoretical prediction.

**Sample efficiency.** We next compare the sample efficiency of the two removal strategies analyzed in Section 3. In Figure 2, we plot removal $\alpha$ as a function of

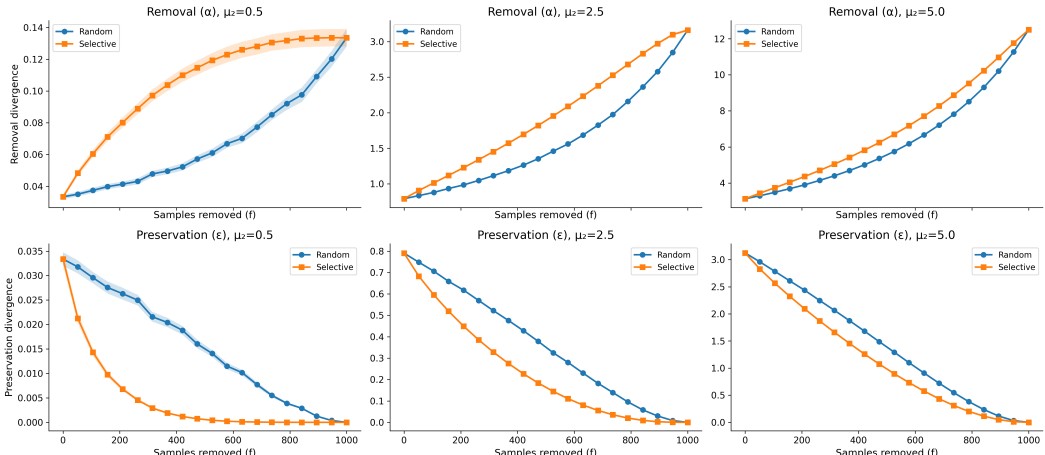

Figure 2: **Synthetic Gaussians.** Selective removal consistently requires fewer deletions, especially when $\mathrm{KL}(p_1\|p_2)$ is small (left), for the same removal and preservation target as random removal. In high-divergence regimes (right), the gap between methods shrinks, as predicted by the theory.

the number of $p_1$ samples removed. Selective removal reaches higher forgetting levels with fewer deletions than random removal, especially in the low-divergence regime ($\mu_2 = 0.5$, top left plot). For example, to reach $0.06$ nats of removal divergence, i.e., half of that obtained by removing all samples, selective removal requires $5\times$ less samples than random removal, i.e., $10\times$ reduction in size from full removal. Analogous trends hold for preservation (bottom left plot): selective removal more effectively preserves the reference distribution $p_2$ throughout. The remaining plots show similar trends when increasing the divergence $\mathrm{KL}(p_1\|p_2)$ by increasing $\mu_2$. As in theory, selective removal offers the greatest savings when $p_1$ and $p_2$ are close and diminishes as the distributions diverge.

## 4.2 CASE STUDIES IN REAL-WORLD DATA

Having validated our theoretical predictions in a controlled setting, we now test our framework's applicability on high-dimensional non-Gaussian data. We present two case studies representing distinct unlearning scenarios. We first analyze the CIFAR-10 dataset (Krizhevsky et al., 2014) to test our framework on well-separated distributions, showing it generalizes standard class unlearning. We then turn to the Jigsaw toxic comments dataset[1] for the more challenging scenario of intertwined distributions, where the unwanted domain is semantically linked to the main task.

**Removal methods.** Across both case studies, we evaluate several scoring heuristics designed to rank samples in the unwanted distribution $p_1$ for selective removal. These heuristics are high-dimensional analogues inspired by the principles developed in our theoretical analysis. The choice of distance metric is adapted to the data modality: for the sparse, high-dimensional TF-IDF text embeddings, we use Cosine distance; for the dense CNN image features, where feature covariance is meaningful, we use Mahalanobis distance[2]. Our main selective removal strategies are: (1) distance to retained (COS-MU2 / MAHA-MU2): this heuristic is a direct analogue of the distance-based method formally analyzed in Section 3. It scores samples based only on their distance to the mean of the retained distribution $p_2$, in cosine and Mahalanobis distance respectively; (2) likelihood-ratio (LR-COS / LR-MAHA): this is an extension that scores samples based on a margin between their distance to the $p_2$ mean versus their distance to the $p_1$ mean, aiming to remove samples that are both distinguishable from $p_2$ and representative of $p_1$, in cosine and Mahalanobis distance respectively; (3) Local Density Ratio (KNN-RATIO): This heuristic estimates the local density ratio around a sample using its $k$-nearest neighbors ($k = 10$). It aims to remove samples that are much more "typical" of the unwanted distribution $p_1$ than the retained distribution $p_2$ in their immediate feature neighborhood; (4) Feature Norm (TFIDF-NORM): This simpler baseline scores samples based on the $\ell_2$-norm of their feature vector. It serves as a proxy for a sample's "informativeness" or extremity in the feature space. We further discuss their computational complexity in Remark 12. While our selection rules follow directly from the Gaussian analysis, we note that real-world distributions can be multimodal, in which case more expressive criteria, e.g., incorporating local density or cluster structure, may further improve performance.

---

[1]https://www.kaggle.com/competitions/jigsaw-toxic-comment-classification-challenge

[2]The Mahalanobis distance of vector $x$ to probability distribution $p$, of mean $\mu$ and covariance $\Sigma$, is: $d(x,p) := \sqrt{(x-\mu)^\top \Sigma^{-1} (x-\mu)}$. We estimate $\mu$ and $\Sigma$ empirically on the retained distribution $p_2$.

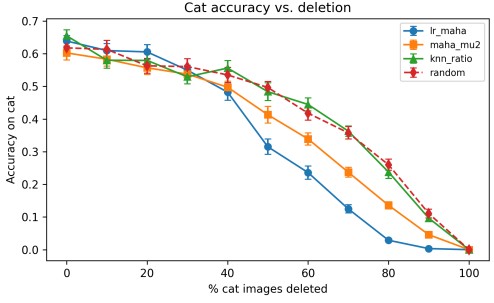 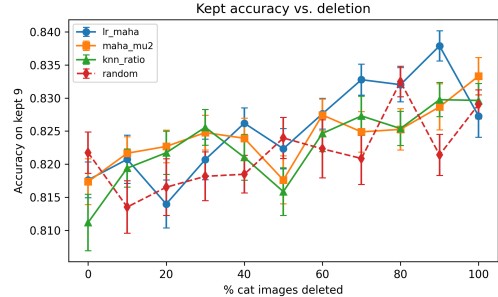

(a) Accuracy on cat images ($p_1$, forgotten).   (b) Accuracy on non-cat images ($p_2$, retained).

Figure 3: **CIFAR-10 images.** Removing cat images suppresses accuracy on that class (left) while leaving accuracy on the retained nine classes essentially unchanged (right, $<0.03$ variation). No substantial removal is observed until $50\%$ deletion, before selective removal strategies LR-MAHA and MAHA-MU2 outperform random removal. Error bars: $\pm1$ standard error over thirty seeds.

Our scoring functions compute similarity between forget and retain samples using finite-sample estimates, e.g., empirical means or embeddings. When only a small or proxy retain subset is available, as may occur in large-scale settings, these estimates become noisier but still supply a meaningful signal, and the distributional unlearning framework applies without modification. Extending these ideas to settings such as large language models, where the retain distribution may be approximated by a general-capabilities corpus, is a natural direction for future work.

**Remark 2** (Selection Budget Choice). *In practical deployments, the deletion budget can be chosen in two complementary ways. First, budget may be constrained by the computational cost of a downstream sample-level unlearning method, in which case one simply targets the largest forget-set size that satisfies this constraint. Second, one may inspect the cumulative distribution of selection scores and choose the smallest subset of samples that accounts for a desired fraction of the total influence (e.g., 80%), analogous to coverage thresholds used in data pruning.*

### 4.2.1 CIFAR-10: VALIDATION ON WELL-SEPARATED DISTRIBUTIONS

We first validate our framework on well-separated distributions using the CIFAR-10 dataset, treating the 'cat' class as the unwanted domain to simulate common class-level unlearning tasks. This allows us to test a key hypothesis: that even within a single, well-defined class, the statistical influence distinguishing it from other classes is not uniformly distributed among its members. To test this, we rank all 'cat' images based on distance scores computed in the feature space of a CNN model, aiming to find the subset whose removal most efficiently erases the class's statistical footprint. We delete the top score-ranked samples for each deletion budget, re-train a convolutional neural network for ten epochs, and report accuracy on the cat test set and accuracy on the other nine classes test set.

**Findings.** The results in Figure 3 confirm our hypothesis. The superior sample efficiency of selective methods shows that even within a single class, statistical influence is concentrated. For instance, the LR-MAHA strategy halves the initial accuracy on the 'cat' class by deleting only $50\%$ of the images, making it $1.6\times$ more data-efficient than random removal. By removing the most statistically distinct cats first, our data-centric approach accelerates the unlearning process while leaving performance on the other nine classes stable (Fig. 3b). This demonstrates our framework's value in class unlearning scenarios, offering a more targeted and efficient alternative to naive or complete removals.

### 4.2.2 JIGSAW TOXIC COMMENTS: INTERTWINED DISTRIBUTION CHALLENGE

We now test our framework on intertwined distributions using the Jigsaw toxic comments dataset. Here, the unwanted domain—comments containing specific profanities ($8.6\%$ of the corpus, chosen keywords in Appendix B)—contains strong predictive signals for the main task of identifying toxicity in the retained, non-profane comments. This creates a high-stakes trade-off where simply removing the entire unwanted $p_1$ samples harms utility, a fact demonstrated by the sharp drop in performance at full deletion in our experiments (Figure 4b). The objective is therefore to remove the influence of explicit profanity while preserving the shared predictive features.

**Findings.** Our results validate the need for a targeted strategy in this setting. Figure 4a shows that recall on profane comments remains high until large deletion budgets are reached, confirming the hardness of the task and suggesting that naively removing random profane comments—which may be statistically similar to non-profane text—is ineffective. A significant unlearning effect is

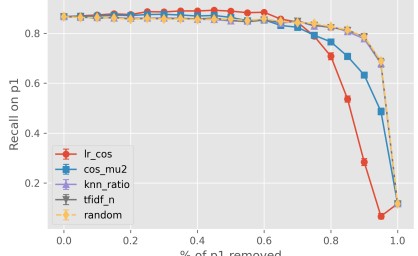 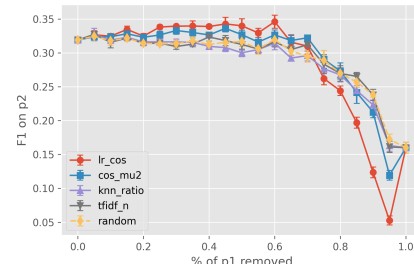

(a) Recall on profane comments ($p_1$, forgotten).  (b) *F1* on non-profane comments ($p_2$, retained).

Figure 4: **Jigsaw Toxic Comments.** Impact of removing profane comments on Jigsaw Toxic. *Left:* recall on the to-be-forgotten set $p_1$; *right:* $F_1$ on the retained set $p_2$. Utility is almost unchanged up to $60\%$ deletion; marked forgetting appears only around $80\%$ deletion, with LR-COS showing the steepest drop. Error bars: $\pm 1$ standard error over five randomness seeds.

only achieved by selective strategies like LR-COS, which prioritize removing outlier, distinguishable comments. For example, to reduce recall on the profane set to 0.70, the LR-COS strategy achieves this same effect by deleting 80%, a 15% reduction in the required deletion budget compared to random removal, while relatively maintaining performance on the retained set unlike complete removal.

## 4.3 SYNERGY WITH SAMPLE-LEVEL UNLEARNING

A key application of our data-centric framework is to serve as an efficiency-boosting front-end for various sample-level unlearning algorithms, as their computational cost typically scales with the size of the flagged forget set (e.g., fewer gradients on forget data, see Remark 11). First, on Gaussian data, we pair our selection method with full retraining and a standard influence function-based approximation using the log-likelihood Hessian. Second, on our more complex CIFAR-10 class unlearning task, we pair with recent methods, including finetuning and advanced, gradient-based techniques like NegGrad+ (Kurmanji et al., 2024) and SalUn (Fan et al., 2024) (App. B.2.2). The results, summarized in Table 5, show a consistent and significant improvement in sample efficiency. This advantage is most pronounced in the synthetic, low-divergence setting where our method is up to $5\times$ more sample-efficient, and it qualitatively generalizes to the CIFAR-10 task, where our method uses a deletion budget up to $2\times$ smaller than full removal, while achieving a strong unlearning effect, i.e., 20% test accuracy on forget data, with original $\approx$86% test accuracy on the retained data. Savings vary following the quality of sample-level unlearning, which is poor for naive finetuning, and much more interesting for SalUn, which approximates the retraining standard well.

We compare against a centroid-based coreset baseline that removes the samples most representative of $p_1$ (those closest to its mean); we also evaluated $k$-center greedy (recalled in App. B.2.1) which performs near-identically. This strategy performs poorly because, in low-divergence settings, it removes exactly the samples that are least distinguishable from $p_2$—the opposite of our selective method. While more sophisticated coreset methods exist, they are designed to preserve the properties of a single distribution. Our dual-distribution objective—to maximize divergence from $p_1$ while minimizing divergence with respect to $p_2$—is fundamentally different. We hypothesize that any method optimizing for within-distribution representativeness will underperform our cross-distributional approach, and leave a deeper adaptation of coreset methods to future work. Indeed, our experiments in Section B.4 with two strong gradient-based coresets (CRAIG (Mirzasoleiman et al., 2020), GradMatch (Killamsetty et al., 2021)) illustrate this limitation empirically, as both outperform random deletion yet fall substantially short of our distributional selection under matched budgets. Similarly, recent pruning methods such as TDDS (Zhang et al., 2024b), CCS (Zheng et al., 2023), and UNSEEN (Xu et al., 2025) further advance data-efficiency for training, but they remain single-distribution methods: their scoring functions are defined entirely on the training distribution. As such, they cannot exploit the contrast between the forget and retain distributions that drives distributional unlearning.

**Remark 3** (Retain data assumption). *When no retain distribution is available, our objective collapses to selecting influential points within the forget distribution, which is more aligned with classical coreset construction. Our experiments indicate that such single-distribution methods are limited in low-divergence regimes: they emphasize representativeness within the forget distribution but cannot leverage the crucial contrast with the retain distribution that drives effective data removal.*

| Unlearning Method | Selection Method | | | Savings |
|---|---|---|---|---|
| | **Random** | **Coreset** | **Selective** | |
| Retraining | 80 | 75 | **60** | **40%** |
| Finetuning | 98 | 98 | **88** | **12%** |
| NegGrad+ | 88 | 85 | **70** | **30%** |
| SalUn | 68 | 75 | **55** | **45%** |

| Selection Method | Unlearning Method | |
|---|---|---|
| | **Retraining** | **Influence Func.** |
| **Selective** | **15** | **18** |
| Random | 61 | 91 |
| Coreset | 63 | 93 |

Table 5: **Synergy with Sample-Level Unlearning.** Deletion budget (in %) required to reach: *(left)* 20% test accuracy on the CIFAR-10 forget class; *(right)* half the initial KL divergence in low-divergence Gaussians (Fig. 2, top left scenario). Our Selective removal is benchmarked against Random selection and a Coreset baseline across various sample-level unlearning methods. "Savings" indicates the relative reduction in size of selective removal from the full forget set.

*For this reason, we view acquiring or defining even a coarse proxy retain dataset (e.g., a general capabilities corpus) as an important practical consideration for applying distributional unlearning.*

## 5 Conclusion and Future Work

Machine unlearning increasingly requires moving beyond individual record deletion to erase the influence of entire subpopulations. We tackled the central dilemma of this task: that full removal is computationally expensive, while naive partial removal is statistically inefficient. We find that a domain's statistical influence is often concentrated in a small high-impact subset of its samples. We formalized this insight as *distributional unlearning*, a framework for selecting a small subset of data that optimally balances forgetting an unwanted distribution while preserving a desired one. Our theoretical analysis provided provable guarantees connecting this data-centric approach to downstream model performance, and our experiments validated that a selective, distance-based removal strategy is often more data-efficient than random or full removals across a range of tasks. While our work provides a foundation for selective data removal, we acknowledge its limitations, which point to important directions for future research below.

**Distributional assumptions.** Our finite-sample analysis provides strong guarantees but assumes Gaussian distributions; while our experiments show the core insights generalize qualitatively, bridging the gap between these theoretical bounds and the behavior on complex real-world data remains an open question. This distributional mismatch also helps explain the gap between the 82% data savings in our low-divergence synthetic setting and the still-significant 15-50% savings on real data. More fundamentally, our framework operates on samples that have already been identified as belonging to an unwanted subpopulation. This leads to an exiciting extension of our work: because we show that only a small subset is needed for effective unlearning, this creates the potential for active identification systems that could help practitioners find these few, high-impact samples at a fraction of the cost of exhaustive annotation.

**Data- to model-level guarantees.** Moreover, our evaluation of unlearning is based on model performance degradation, which directly validates our theoretical log-loss guarantees. A privacy-centric evaluation could employ methods like Membership Inference attacks (Shokri et al., 2017) to formally verify that the unlearned model is indistinguishable from one retrained from scratch, representing another crucial avenue for future work. Conceptually, one could make a formal link between the (sample-level) certified (Guo et al., 2020) and distributional unlearning frameworks.

**LLM fine-tuning applications.** Beyond the foundational discriminative settings studied here, there is now a rapidly expanding line of work on data selection for large language model (LLM) instruction tuning. Classical instruction tuning (Wei et al., 2022), demonstrates that finetuning on curated instruction datasets can significantly improve generalization. More recent approaches focus on selecting high-quality subsets from a single instruction distribution to improve supervised finetuning (SFT) efficiency (Li et al., 2024a;b; Wang et al., 2025); see also the survey of Zhang et al. (2025). These methods aim to upweight instructive or representative examples for SFT, whereas our framework addresses a fundamentally different objective: distributional unlearning requires selecting a deletion subset that shifts a model's behavior between two distributions. In this sense, our theory is model-agnostic and can serve as the selection layer for future (multimodal) LLM unlearning pipelines by operating directly on data representations (e.g., LLM embeddings) and respecting forget/retain distributional structure.

## ACKNOWLEDGEMENTS

YA acknowledges support by SNSF grant 200021_200477 and the G-Research PhD prize. SK acknowledges support by NSF 2046795 and 2205329, IES R305C240046, ARPA-H, the MacArthur Foundation, Schmidt Sciences, HAI, OpenAI, Microsoft, and Google. SK and YA acknowledge support by the Center for AI Safety.

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

# A PROOFS

## A.1 PARETO FRONTIER

**Proposition 1** (Pareto Frontier). *Let $p_1, p_2$ be two distributions in $\mathcal{P}$, the class of Gaussian distributions with shared positive covariance. The Pareto frontier of $(\alpha, \varepsilon)$ values achievable in $\mathcal{P}$ is:*

$$\mathrm{PF}(p_1, p_2; \mathcal{P}) = \left\{ \left( \alpha, \left( \sqrt{\alpha} - \sqrt{\mathrm{KL}(p_1 \| p_2)} \right)^2 \right) : \alpha \geq \mathrm{KL}(p_1 \| p_2) \right\}.$$

*Proof.* For simplicity, we consider univariate Gaussians with shared variance. For $d$-dimensional Gaussians with covariance $\Sigma \in \mathbb{R}^{d \times d}$, the same result holds after replacing squared error by the Mahalanobis distance $\|x - \mu\|_{\Sigma^{-1}}^2$ (see Proposition 4).

Let $p = \mathcal{N}(\mu, \sigma^2) \in \mathcal{P}$. Since all distributions in $\mathcal{P}$ share the same variance, the KL divergence from $p_i$ to $p$ is:

$$\mathrm{KL}(p_i \| p) = \frac{(\mu_i - \mu)^2}{2\sigma^2}, \quad i = 1, 2.$$

Fix $\alpha \geq \mathrm{KL}(p_1 \| p_2)$. We want to compute the minimal possible $\varepsilon$ achievable under the constraint $\mathrm{KL}(p_1 \| p) \geq \alpha$. Define this minimum as:

$$\varepsilon_\star(\alpha) := \min_{\mu \in \mathbb{R}, \ (\mu - \mu_1)^2 \geq 2\sigma^2 \alpha} \frac{(\mu_2 - \mu)^2}{2\sigma^2}.$$

This is a one-dimensional quadratic minimization problem subject to a quadratic inequality constraint. The feasible set is:

$$\mu \in (-\infty, \mu_1 - \sigma\sqrt{2\alpha}] \cup [\mu_1 + \sigma\sqrt{2\alpha}, \infty).$$

We minimize $(\mu_2 - \mu)^2$ over this set. This yields two cases:

- If $\mu_2 \in [\mu_1 - \sigma\sqrt{2\alpha}, \mu_1 + \sigma\sqrt{2\alpha}]$, then the closest feasible points are the endpoints. The minimizing value of $\mu$ is:

$$\mu = \mu_1 + \mathrm{sign}(\mu_2 - \mu_1) \cdot \sigma\sqrt{2\alpha},$$

  and the resulting divergence is:

$$\varepsilon_\star(\alpha) = \frac{(\mu_2 - \mu_1 - \mathrm{sign}(\mu_2 - \mu_1) \cdot \sigma\sqrt{2\alpha})^2}{2\sigma^2}.$$

- If $\mu_2$ already lies in the feasible set, i.e., $|\mu_2 - \mu_1| \geq \sigma\sqrt{2\alpha}$, then we can choose $\mu = \mu_2$, yielding $\varepsilon_\star(\alpha) = 0$.

Thus, for all $\alpha \geq 0$:

$$\varepsilon_\star(\alpha) = \frac{\left[ \left( |\mu_2 - \mu_1| - \sigma\sqrt{2\alpha} \right)_+ \right]^2}{2\sigma^2},$$

where $(x)_+ = \max\{x, 0\}$. Let $\Delta := |\mu_2 - \mu_1|$, and recall that $\mathrm{KL}(p_1 \| p_2) = \frac{\Delta^2}{2\sigma^2}$. Then:

$$\Delta = \sigma\sqrt{2\mathrm{KL}(p_1 \| p_2)}.$$

Substituting into $\varepsilon_\star(\alpha)$:

$$\varepsilon_\star(\alpha) = \left( \sqrt{\alpha} - \sqrt{\mathrm{KL}(p_1 \| p_2)} \right)^2, \quad \text{for } \alpha \geq \mathrm{KL}(p_1 \| p_2).$$

Finally, note that any pair $(\alpha, \varepsilon)$ with $\alpha < \mathrm{KL}(p_1 \| p_2)$ satisfies $\varepsilon_\star(\alpha) = 0$, hence is dominated by $(\mathrm{KL}(p_1 \| p_2), 0)$. Therefore, the Pareto optimal points are exactly:

$$\left\{ \left( \alpha, \left( \sqrt{\alpha} - \sqrt{\mathrm{KL}(p_1 \| p_2)} \right)^2 \right) : \alpha \geq \mathrm{KL}(p_1 \| p_2) \right\},$$

as claimed. $\qquad\square$

Next, we show that a qualitatively similar result holds more generally for any exponential-family member.

**Theorem 4** (Pareto Frontier–Exponential Families). *Let $(\mathcal{X}, \mu)$ be a measurable space and let*

$$\mathcal{P} = \left\{ p_\theta(x) = h(x) \exp(\theta^\top T(x) - A(\theta)) : \theta \in \Theta \subset \mathbb{R}^d \right\}$$

*be a regular minimal exponential family (carrier $h > 0$, sufficient statistic $T \colon \mathcal{X} \to \mathbb{R}^d$, log-partition $A$). Fix $p_i(x) = p_{\theta_i}(x)$, $i = 1, 2$, and $\alpha \geq 0$. Define $v(\alpha) = \inf_{\theta \in \Theta} \{ \mathrm{KL}(p_2 \| p_\theta) \mid \mathrm{KL}(p_1 \| p_\theta) \geq \alpha \}$, where $\mathrm{KL}(q \| p) = \int q \log(q/p) \, d\mu$. Then:*

*(i) The Pareto frontier for points in $\mathcal{P}$ is*

$$\mathrm{PF}(p_1, p_2; \mathcal{P}) = \left\{ (\alpha, v(\alpha)) : \alpha \geq \mathrm{KL}(p_1 \| p_2) \right\},$$

*where $v(\alpha) = \mathrm{KL}(p_2 \| p_1) + \alpha + \frac{1}{\lambda^* - 1} (\theta_2 - \theta_1)^\top (\mathbb{E}_{p_2}[T] - \mathbb{E}_{p_1}[T])$, and $\lambda^*$ is the unique scalar in $(0, 1)$ such that the distribution $p^* \in \mathcal{P}$ of mean $\mathbb{E}_{p^*}[T] = \frac{\lambda^* \mathbb{E}_{p_1}[T] - \mathbb{E}_{p_2}[T]}{\lambda^* - 1}$ satisfies $\mathrm{KL}(p_1 \| p^*) = \alpha$.*

*(ii) **(Gaussian case)** If $\mathcal{P} = \{\mathcal{N}(\mu, \Sigma) : \mu \in \mathbb{R}^d\}$ with fixed $\Sigma \succ 0$, then for $\alpha \geq \mathrm{KL}(p_1 \| p_2)$:*

$$v(\alpha) = \left( \sqrt{\alpha} - \sqrt{\mathrm{KL}(p_1 \| p_2)} \right)^2.$$

*Proof.* In this proof, for a fixed $\alpha \geq 0$, we study the optimal value

$$v(\alpha) = \inf_{\theta \in \Theta} \{ \mathrm{KL}(p_2 \| p_\theta) \mid \mathrm{KL}(p_1 \| p_\theta) \geq \alpha \}.$$

This enables characterzing the Pareto frontier of feasible $(\alpha, \varepsilon)$ trade-offs. Below, we focus on the non-trivial case $\alpha > \mathrm{KL}(p_1 \| p_2)$. Indeed, in the case $\alpha \leq \mathrm{KL}(p_1 \| p_2)$, the problem above's unconstrained minimizer $p_2$ is a feasible solution, so that $v(\alpha) = 0$ for all $\alpha \leq \mathrm{KL}(p_1 \| p_2)$.

**Reparametrization and KKT.** First, we recall the expression of the KL divergence in exponential families as a Bregman divergence. For any $\theta', \theta \in \Theta$ one has

$$\mathrm{KL}(p_{\theta'} \| p_\theta) = A(\theta) - A(\theta') - (\theta - \theta')^T \nabla A(\theta').$$

We let

$$f(\theta) = \mathrm{KL}(p_2 \| p_\theta), \quad g(\theta) = \mathrm{KL}(p_1 \| p_\theta).$$

Both are continuously differentiable on the open set $\Theta$. The feasible set $\{g(\theta) \geq \alpha\}$ is nonconvex, so we check linear independence constraint qualification (LICQ, Bertsekas (1997)) at any minimizer $\theta^*$. To do so, we recall (see Wainwright et al. (2008)) that for exponential families $\nabla A(\theta) = \mathbb{E}_{p_\theta}[T]$ for any $\theta \in \Theta$, so that

$$\nabla g(\theta) = -\nabla A(\theta_1) + \nabla A(\theta) = -\mathbb{E}_{p_1}[T] + \mathbb{E}_{p_\theta}[T].$$

Minimality of the exponential family ensures $\mathbb{E}_{p_\theta}[T]$ is injective Wainwright et al. (2008). This together with the fact that $p_1 \neq p_\theta$ for any feasible $\theta$, since $\alpha > 0$, implies that $\nabla g(\theta^*) \neq 0$ and LICQ holds.

Next, we form the Lagrangian

$$\mathcal{L}(\theta, \lambda) = f(\theta) - \lambda \big( g(\theta) - \alpha \big), \quad \lambda \geq 0.$$

Since LICQ holds, KKT conditions are necessary Bertsekas (1997) for any minimizer $\theta^*$. Hence, stationarity ($\nabla_\theta \mathcal{L} = 0$) gives

$$\nabla f(\theta^*) = \lambda \nabla g(\theta^*).$$

Given that we had derived $\nabla_\theta \mathrm{KL}(p_i \| p_\theta) = -\mathbb{E}_{p_i}[T] + \mathbb{E}_{p_\theta}[T]$, we obtain

$$-\mathbb{E}_{p_2}[T] + \mathbb{E}_{p^*}[T] = \lambda \left( -\mathbb{E}_{p_1}[T] + \mathbb{E}_{p^*}[T] \right),$$

hence $(1 - \lambda) \mathbb{E}_{p^*}[T] = \mathbb{E}_{p_2}[T] - \lambda \mathbb{E}_{p_1}[T]$. Observe that we must have $\lambda \neq 1$, as otherwise $\mathbb{E}_{p_2}[T] = \mathbb{E}_{p_1}[T]$, but this contradicts the fact that $p_1 \neq p_1$ given that $\mathbb{E}_{p_\theta}[T]$ is injective for minimal exponential families Wainwright et al. (2008). Therefore, we have the following:

$$\mathbb{E}_{p^*}[T] = \frac{\lambda \mathbb{E}_{p_1}[T] - \mathbb{E}_{p_2}[T]}{\lambda - 1}.$$

Now, we observe that the inequality constraint must be active. Otherwise, the minimizer lies in the interior of the feasible set and is thus a local minimum of the unconstrained problem. The latter admits $p_2$ as a unique minimizer, so the minimizer at hand must be $p_2$ but this is not feasible since we assume $\mathrm{KL}(p_1\|p_2) < \alpha$. Therefore, the minimizer must lie at the boundary of the feasible set, and the inequality constraint is active. That is, we have $g(\theta^*) = \alpha$.

**Optimal value.** We are now ready to derive the expression of the optimal value:

$$v(\alpha) = f(\theta^\star) = \mathrm{KL}(p_2\|p^*).$$

We use the following classical Pythagorean-type identity for Bregman divergences (see, e.g., Banerjee et al. Banerjee et al. (2005)):

$$\mathrm{KL}(p_2\|p^*) = \mathrm{KL}(p_2\|p_1) + \mathrm{KL}(p_1\|p^*) + (\theta_2 - \theta_1)^\top (\nabla A(\theta_1) - \nabla A(\theta^*)).$$

Now, consider the aforementioned KKT multiplier $\lambda > 0, \lambda \neq 1$ such that $\nabla A(\theta^*) = \mathbb{E}_{p^*}[T] = \frac{\lambda \mathbb{E}_{p_1}[T] - \mathbb{E}_{p_2}[T]}{\lambda - 1} = \frac{\lambda \nabla A(\theta_1) - \nabla A(\theta_2)}{\lambda - 1}$ as well as $\mathrm{KL}(p_1\|p^*) = g(\theta^*) = \alpha$. We thus get

$$v(\alpha) = \mathrm{KL}(p_2\|p^*) = \mathrm{KL}(p_2\|p_1) + \mathrm{KL}(p_1\|p^*) + (\theta_2 - \theta_1)^\top (\nabla A(\theta_1) - \nabla A(\theta^*))$$

$$= \mathrm{KL}(p_2\|p_1) + \mathrm{KL}(p_1\|p^*) + \frac{1}{\lambda - 1}(\theta_2 - \theta_1)^\top (\nabla A(\theta_2) - \nabla A(\theta_1))$$

$$= \mathrm{KL}(p_2\|p_1) + \alpha + \frac{1}{\lambda - 1}(\theta_2 - \theta_1)^\top (\nabla A(\theta_2) - \nabla A(\theta_1)).$$

We now again use that $\nabla A(\theta) = \mathbb{E}_{p_\theta}[T]$ for all $\theta \in \Theta$. We then obtain:

$$v(\alpha) = \mathrm{KL}(p_2\|p_1) + \alpha + \frac{1}{\lambda - 1}(\theta_2 - \theta_1)^\top (\mathbb{E}_{p_2}[T] - \mathbb{E}_{p_1}[T]). \tag{3}$$

**Uniqueness of $\lambda$.** We now show that there is a unique KKT multiplier $\lambda$ for the optimal solution. We recall that $\lambda > 0$ is such that $\lambda \neq 1$ and:

$$\mathbb{E}_{p^*}[T] = \frac{\lambda \mathbb{E}_{p_1}[T] - \mathbb{E}_{p_2}[T]}{\lambda - 1} \qquad\qquad g(\theta^*) = \mathrm{KL}(p_1\|p^*) = \alpha.$$

Above, the first equation uniquely defines the distribution $p^*$, by minimality of the family, and we now show that there is only a unique $\lambda$ of interest such that the second equations above holds.

Let $\theta^*(\lambda)$ be the unique (by minimality) parameter such that $\mathbb{E}_{p^*}[T] = \frac{\lambda \mathbb{E}_{p_1}[T] - \mathbb{E}_{p_2}[T]}{\lambda - 1}$. We define

$$H(\lambda) := \mathrm{KL}(p_1\|p^*) = A(\theta^*) - A(\theta_1) - (\theta^* - \theta_1)^\top \nabla A(\theta_1)$$

Taking the derivative above, we get

$$\frac{dH}{d\lambda}(\lambda) = (\nabla A(\theta^*) - \nabla A(\theta_1))^\top \frac{d\theta^*}{d\lambda}(\lambda).$$

We again use that

$$\nabla A(\theta^*) = \mathbb{E}_{p^*}[T] = \frac{\lambda \mathbb{E}_{p_1}[T] - \mathbb{E}_{p_2}[T]}{\lambda - 1} = \frac{\lambda \nabla A(\theta_1) - \nabla A(\theta_2)}{\lambda - 1}.$$

First, replacing in the previous derivative equation, we obtain:

$$\frac{dH}{d\lambda}(\lambda) = (\nabla A(\theta^*) - \nabla A(\theta_1))^\top \frac{d\theta^*}{d\lambda}(\lambda) = \frac{1}{\lambda - 1}(\nabla A(\theta_1) - \nabla A(\theta_2))^\top \frac{d\theta^*}{d\lambda}(\lambda).$$

Second, taking the derivative with respect to $\lambda$ in the expression of $\nabla A(\theta^*)$ yields:

$$\nabla^2 A(\theta^*) \cdot \frac{d\theta^*}{d\lambda}(\lambda) = \frac{1}{(\lambda - 1)^2}(\nabla A(\theta_1) - \nabla A(\theta_2)).$$

We observe that the Fisher information matrix $\nabla^2 A(\theta^*)$ is positive definite since the family is regular. Multiplying by the inverse of the latter then yields:

$$\frac{d\theta^*}{d\lambda}(\lambda) = \frac{1}{(\lambda - 1)^2}\nabla^2 A(\theta^*)^{-1}(\nabla A(\theta_1) - \nabla A(\theta_2)).$$

Plugging the above in the latest expression of the derivative of $H$ yields:

$$\frac{dH}{d\lambda}(\lambda) = \frac{1}{\lambda - 1}(\nabla A(\theta_1) - \nabla A(\theta_2))^\top \frac{d\theta^*}{d\lambda}(\lambda)$$

$$= \frac{1}{(\lambda - 1)^3}(\nabla A(\theta_1) - \nabla A(\theta_2))^\top \nabla^2 A(\theta^*)^{-1}(\nabla A(\theta_1) - \nabla A(\theta_2)).$$

Since the matrix $\nabla^2 A(\theta^*)$ is positive definite by regularity of the family, the corresponding quadratic form above is positive (recall $\theta_1 \neq \theta_2$), and the sign of the derivative is that of $\lambda - 1$. Therefore, $H$ is decreasing on $(0, 1)$ and increasing on $(1, +\infty)$. It is straighforward to check that $H(0^+) = \mathrm{KL}(p_1 \| p_2)$, $H(1^-) = H(1^+) = +\infty$, and $H(+\infty) = 0$. Since $\mathrm{KL}(p_1 \| p_2) < \alpha$ by assumption, there exists a unique $\lambda^* \in (0, 1)$ such that $H(\lambda^*) = \alpha$ and a unique $\lambda_1^* > 1$ such that $H(\lambda_1^*) = \alpha$.

We now discard $\lambda_1^*$ thanks to the expression of the optimal value expression (3). Indeed, the second term in (3) is positive for $\lambda_1^*$ since $\lambda_1^* > 1$ and $(\theta_2 - \theta_1)^\top (\mathbb{E}_{p_2}[T] - \mathbb{E}_{p_1}[T]) = (\theta_2 - \theta_1)^\top (\nabla A(\theta_2) - \nabla A(\theta_1)) > 0$ by strict convexity of $A$ and the fact that $p_1 \neq p_2$. On the other hand, this same second term is negative for $\lambda_*$ since $\lambda_* < 1$. Therefore, the optimal value is smaller for the choice of $\lambda^*$, so that:

$$v(\alpha) = \mathrm{KL}(p_2 \| p_1) + \alpha + \frac{1}{\lambda^* - 1}(\theta_2 - \theta_1)^\top (\mathbb{E}_{p_2}[T] - \mathbb{E}_{p_1}[T]), \tag{4}$$

where $\lambda^*$ is the unique scalar in $(0, 1)$ such that:

$$\mathbb{E}_{p^*}[T] = \frac{\lambda^* \mathbb{E}_{p_1}[T] - \mathbb{E}_{p_2}[T]}{\lambda^* - 1} \qquad g(\theta^*) = \mathrm{KL}(p_1 \| p^*) = \alpha.$$

Finally, we note that $v(\alpha)$ is non-decreasing by definition; increasing $\alpha$ shrinks the feasible set. Also, we recall that $v(\alpha) = 0$ for all $\alpha < \mathrm{KL}(p_1 \| p_2)$, i.e., all trade-offs $(\alpha, \varepsilon)$ with $\alpha < \mathrm{KL}(p_1 \| p_2), \varepsilon > 0$ are dominated by $(\mathrm{KL}(p_1 \| p_2), 0)$. Therefore, we conclude that the pareto frontier is given by:

$$\mathrm{PF}(p_1, p_2; \mathcal{P}) = \Big\{ (\alpha, v(\alpha)) : \alpha \geq \mathrm{KL}(p_1 \| p_2) \Big\}.$$

**Gaussian case.** For $p_\mu = \mathcal{N}(\mu, \Sigma)$ one has $T(x) = x$, $E_{p_\mu}[T] = \mu$, and

$$\mathrm{KL}(p_i \| p_\mu) = \tfrac{1}{2}(\mu_i - \mu)^\top \Sigma^{-1}(\mu_i - \mu).$$

By the conditions on $\lambda^*$ we have

$$\mu^* = \frac{\lambda^* \mu_1 - \mu_2}{\lambda^* - 1}, \quad \mathrm{KL}(p_1 \| p_{\mu^*}) = \alpha.$$

Using the KL divergence expression for Gaussians $\mathcal{N}(\mu, \Sigma)$ (recall $\mathrm{KL}(\mathcal{N}(\mu_1, \Sigma), \mathcal{N}(\mu_2, \Sigma)) = \tfrac{1}{2}(\mu_1 - \mu_2)^\top \Sigma^{-1}(\mu_1 - \mu_2))$, we get

$$\mu_1 - \mu^* = \frac{\mu_2 - \mu_1}{\lambda^* - 1}, \quad \alpha = \frac{\mathrm{KL}(p_1 \| p_2)}{(\lambda^* - 1)^2}.$$

Solving for $\lambda^* \in (0, 1)$ yields $\lambda^* = 1 - \sqrt{\mathrm{KL}(p_1 \| p_2)/\alpha}$ and

$$\mu^* = \mu_1 + \sqrt{\frac{\alpha}{\mathrm{KL}(p_1 \| p_2)}}(\mu_2 - \mu_1).$$

Thus, direct computations yield

$$v(\alpha) = \tfrac{1}{2}\Big( \|\mu_2 - \mu_1\|_{\Sigma^{-1}} - \sqrt{2\alpha} \Big)^2 = \big( \sqrt{\mathrm{KL}(p_1 \| p_2)} - \sqrt{\alpha} \big)^2,$$

with $v(\alpha) = 0$ if $\alpha \leq \mathrm{KL}(p_1 \| p_2)$. This concludes the proof. $\qquad \square$

**Discussion.** In Proposition 4 we show that, in any regular exponential family, the trade-off between removal ($\alpha$) and preservation ($\varepsilon$) can be quantified. This yields a removal-preservation trade-off curve that faithfully reproduces the shared-covariance Gaussian Pareto frontier—namely the familiar $(\sqrt{\alpha} - \sqrt{D})^2$ parabola—while in other families it gives an explicit but generally non-algebraic trade-off curve.

A.2    PREDICTIVE PERFORMANCE

**Proposition 2.** *Let $h$ minimize $\mathcal{L}(h; p)$, and let $h_1$, $h_2$ be optimal predictors under $p_1, p_2 \in \mathcal{P}$, respectively. If $p$ satisfies $(\alpha, \varepsilon)$-distributional unlearning with respect to $(p_1, p_2)$, then:*

$$\mathcal{L}(h; p_1) - \mathcal{L}(h_1; p_1) \geq \alpha - \delta_1, \qquad\qquad \mathcal{L}(h; p_2) - \mathcal{L}(h_2; p_2) \leq \varepsilon - \delta_2, \qquad (2)$$

*where $\delta_1 := \mathrm{KL}(p_{1,X} \| p_X), \delta_2 := \mathrm{KL}(p_{2,X} \| p_X)$ denote the marginal KL divergence over inputs.*

*Proof.* Let $h(y \mid x) := h(x)(y)$ denote the conditional distribution defined by the hypothesis $h$, and suppose $h$ minimizes the expected log-loss under $p$. Since $\ell(y, q) = -\log q(y)$ is a strictly proper scoring rule, the unique minimizer of $\mathcal{L}(h; p)$ is the true conditional distribution $h(x) = p(\cdot \mid x)$, where $p(x, y) = p^X(x)p(y \mid x)$.

We begin by analyzing the expected log-loss of this hypothesis under an arbitrary distribution $q$ over $\mathcal{X} \times \mathcal{Y}$:

$$\mathcal{L}(h; q) = \mathbb{E}_{(x,y) \sim q}[-\log h(y \mid x)] = \mathbb{E}_{x \sim q^X} \mathbb{E}_{y \sim q(\cdot \mid x)}[-\log p(y \mid x)],$$

where $q^X$ denotes the marginal distribution of $x$ under $q$, and $q(\cdot \mid x)$ the corresponding conditional.

Now recall the standard identity for any two conditional distributions $q(\cdot \mid x)$ and $p(\cdot \mid x)$:

$$\mathbb{E}_{y \sim q(\cdot \mid x)}[-\log p(y \mid x)] = \mathrm{KL}(q(y \mid x) \| p(y \mid x)) + H(q(y \mid x)),$$

where $H(q(y \mid x)) = \mathbb{E}_{y \sim q(\cdot \mid x)}[-\log q(y \mid x)]$ is the Shannon entropy of the label distribution under $q$ for fixed $x$.

Taking the expectation over $x \sim q^X$, we get:

$$\begin{aligned}
\mathcal{L}(h; q) &= \mathbb{E}_{x \sim q^X}\left[\mathrm{KL}(q(y \mid x) \| p(y \mid x)) + H(q(y \mid x))\right] \\
&= \mathbb{E}_{x \sim q^X} \mathrm{KL}(q(y \mid x) \| p(y \mid x)) + \mathbb{E}_{x \sim q^X} H(q(y \mid x)).
\end{aligned}$$

The second term is the expected entropy, which corresponds to the Bayes-optimal risk under $q$:

$$\mathcal{L}(h_q^*; q) := \inf_{h'} \mathcal{L}(h'; q) = \mathbb{E}_{x \sim q^X} H(q(y \mid x)). \qquad (5)$$

Next, we relate the expected conditional KL term to the total KL divergence between the joint distributions. Using the chain rule for KL divergence, we have:

$$\mathrm{KL}(q \| p) = \mathrm{KL}(q^X \| p^X) + \mathbb{E}_{x \sim q^X} \mathrm{KL}(q(y \mid x) \| p(y \mid x)). \qquad (6)$$

This decomposition holds generally for joint distributions with conditional factorizations. Solving for the conditional KL term, we obtain:

$$\mathbb{E}_{x \sim q^X} \mathrm{KL}(q(y \mid x) \| p(y \mid x)) = \mathrm{KL}(q \| p) - \mathrm{KL}(q^X \| p^X). \qquad (7)$$

Substituting the above into the expression for $\mathcal{L}(h; q)$ and using (5), we get:

$$\mathcal{L}(h; q) = \mathrm{KL}(q \| p) - \mathrm{KL}(q^X \| p^X) + \mathcal{L}(h_q^*; q). \qquad (8)$$

We now apply this to $q = p_1$ and $q = p_2$, noting that $p$ satisfies $(\alpha, \varepsilon)$-distributional unlearning, i.e., $\mathrm{KL}(p_1 \| p) \geq \alpha$ and $\mathrm{KL}(p_2 \| p) \leq \varepsilon$.

For $p_1$, we define $\delta_1 := \mathrm{KL}(p_1^X \| p^X)$ and compute:

$$\mathcal{L}(h; p_1) - \mathcal{L}(h_1; p_1) = \mathrm{KL}(p_1 \| p) - \mathrm{KL}(p_1^X \| p^X) = \mathrm{KL}(p_1 \| p) - \delta_1 \geq \alpha - \delta_1.$$

For $p_2$, define $\delta_2 := \mathrm{KL}(p_2^X \| p^X)$ and similarly compute:

$$\mathcal{L}(h; p_2) - \mathcal{L}(h_2; p_2) = \mathrm{KL}(p_2 \| p) - \mathrm{KL}(p_2^X \| p^X) = \mathrm{KL}(p_2 \| p) - \delta_2 \leq \varepsilon - \delta_2.$$

This completes the proof. $\qquad\square$

## A.3 RANDOM REMOVAL

**Lemma 5** (Finite-sample concentration). *Let $\hat{\mu}$ be the empirical mean of $n$ samples drawn from $\mathcal{N}(\mu, \sigma^2)$. For any $\delta \in (0, 1)$, with probability at least $1 - \delta$, we have*

$$|\hat{\mu} - \mu| \leq \sigma\sqrt{\frac{2\ln(2/\delta)}{n}}.$$

*Proof.* This follows directly from Hoeffding's inequality for sub-Gaussian variables. $\square$

**Proposition 3** (Random Removal). *Let $p_1, p_2 \in \mathcal{P}$ and $\delta \in (0, 1)$. We observe $n_1$ samples from $p_1$ and $n_2$ samples from $p_2$, and randomly remove $f$ samples from $p_1$ before fitting. With probability $1 - \delta$, the resulting MLE distribution satisfies $(\alpha, \varepsilon)$-distributional unlearning with:*

$$\alpha \geq \left(\frac{1}{2} - 3\left(\frac{n_1 - f}{n_2}\right)^2\right) \mathrm{KL}(p_1 \parallel p_2) - \frac{3\ln(4/\delta)}{2n_2}\left(1 + \frac{n_1 - f}{n_2}\right),$$

$$\varepsilon \leq 3\left(\frac{n_1 - f}{n_2}\right)^2 \mathrm{KL}(p_1 \parallel p_2) + \frac{3\ln(4/\delta)}{n_2}\left(1 + \frac{n_1 - f}{n_2}\right).$$

*Proof.* We recall that $p_1 = \mathcal{N}(\mu_1, \sigma^2), p_2 = \mathcal{N}(\mu_2, \sigma^2) \in \mathcal{P}$ and $p := \mathcal{N}(\mu, \sigma^2) \in \mathcal{P}$ are univariate Gaussian distributions. We are given $n_1$ i.i.d. samples $x_1^{(1)}, \ldots, x_1^{(n_1)}$ from $p_1$ and $n_2$ i.i.d. samples $x_2^{(1)}, \ldots, x_2^{(n_2)}$ from $p_2$.

Upon removing $f \leq n_1$ randomly chosen samples $x_1^{(1)}, \ldots, x_1^{(n_1 - f)}$ from the target distribution $p_1$, we set the center $\mu$ of the unlearned distribution $p$ to be:

$$\mu = \frac{(n_1 - f)\hat{\mu}_1 + n_2\hat{\mu}_2}{n_1 - f + n_2}, \tag{9}$$

where $\hat{\mu}_1 := \frac{1}{n_1 - f}\sum_{i=1}^{n_1 - f} x_1^{(i)}$ and $\hat{\mu}_2 := \frac{1}{n_2}\sum_{i=1}^{n_2} x_2^{(i)}$. We also observe that a standard Hoeffding bound (Lemma 5) yields that:

$$|\hat{\mu}_1 - \mu_1| \leq \sigma\sqrt{\frac{2\ln(4/\delta)}{f}}, \quad |\hat{\mu}_2 - \mu_2| \leq \sigma\sqrt{\frac{2\ln(4/\delta)}{n_2}}, \tag{10}$$

each with probability $1 - \frac{\delta}{2}$, so that both hold with probability $1 - \delta$ thanks to a union bound. We also recall that

$$\mathrm{KL}(p_1 \parallel p) = \frac{(\mu_1 - \mu)^2}{2\sigma^2}, \quad \mathrm{KL}(p_2 \parallel p) = \frac{(\mu_2 - \mu)^2}{2\sigma^2}. \tag{11}$$

**Preservation bound.** First, we upper bound the KL divergence of $p_2$ from $p$. To do so, we first use the triangle inequality to get

$$|\mu - \mu_2| = \left|\frac{(n_1 - f)\hat{\mu}_1 + n_2\hat{\mu}_2}{n_1 - f + n_2} - \mu_2\right| = \left|\frac{n_1 - f}{n_1 - f + n_2}(\hat{\mu}_1 - \mu_2) + \frac{n_2}{n_1 - f + n_2}(\hat{\mu}_2 - \mu_2)\right|$$

$$= \left|\frac{n_1 - f}{n_1 - f + n_2}(\mu_1 - \mu_2) + \frac{n_1 - f}{n_1 - f + n_2}(\hat{\mu}_1 - \mu_1) + \frac{n_2}{n_1 - f + n_2}(\hat{\mu}_2 - \mu_2)\right|$$

$$\leq \frac{n_1 - f}{n_1 - f + n_2}|\mu_1 - \mu_2| + \frac{n_1 - f}{n_1 - f + n_2}|\hat{\mu}_1 - \mu_1| + \frac{n_2}{n_1 - f + n_2}|\hat{\mu}_2 - \mu_2|.$$

Therefore, using (10) we have with probability $1 - \delta$:

$$|\mu - \mu_2| \leq \frac{n_1 - f}{n_1 - f + n_2}|\mu_1 - \mu_2| + \frac{n_1 - f}{n_1 - f + n_2}\sigma\sqrt{\frac{2\ln(4/\delta)}{f}} + \frac{n_2}{n_1 - f + n_2}\sigma\sqrt{\frac{2\ln(4/\delta)}{n_2}}.$$

Taking squares, using Jensen's inequality, and simplifying further since $f \geq 0$, yields:

$$|\mu - \mu_2|^2 \leq 3\left(\frac{n_1 - f}{n_2}\right)^2|\mu_1 - \mu_2|^2 + 3\left(\frac{n_1 - f}{n_2}\right)^2\sigma^2\frac{2\ln(4/\delta)}{n_1 - f} + \sigma^2\frac{6\ln(4/\delta)}{n_2}. \tag{12}$$

Dividing both sides by $2\sigma^2$ and then using (28) yields with probability $1 - \delta$:

$$\text{KL}(p_2 \parallel p) \leq 3 \left( \frac{n_1 - f}{n_2} \right)^2 \text{KL}(p_1 \parallel p_2) + \frac{3 \ln(4/\delta)}{n_2} \left( 1 + \frac{n_1 - f}{n_2} \right). \tag{13}$$

**Removal bound.** Second, we lower bound the KL divergence of $p_1$ from $p$. To do so, we use Jensen's inequality and (12) to obtain that, with probability $1 - \delta$, we have

$$|\mu_1 - \mu_2|^2 = |\mu_1 - \mu + \mu - \mu_2|^2 \leq 2|\mu_1 - \mu|^2 + 2|\mu - \mu_2|^2$$

$$\leq 2|\mu_1 - \mu|^2 + 6 \left( \frac{n_1 - f}{n_2} \right)^2 |\mu_1 - \mu_2|^2 + \frac{6\sigma^2 \ln(4/\delta)}{n_2} \left( 1 + \frac{n_1 - f}{n_2} \right)$$

Rearranging terms and dividing by $4\sigma^2$ along with (28) yields that with probability $1 - \delta$ we have

$$\text{KL}(p_1 \parallel p) = \frac{|\mu_1 - \mu|^2}{2\sigma^2} \geq \frac{|\mu_1 - \mu_2|^2}{4\sigma^2} - 3 \left( \frac{n_1 - f}{n_2} \right)^2 \frac{|\mu_1 - \mu_2|^2}{2\sigma^2} - \frac{3 \ln(4/\delta)}{2n_2} \left( 1 + \frac{n_1 - f}{n_2} \right)$$

$$= \left( \frac{1}{2} - 3 \left( \frac{n_1 - f}{n_2} \right)^2 \right) \text{KL}(p_1 \parallel p_2) - \frac{3 \ln(4/\delta)}{2n_2} \left( 1 + \frac{n_1 - f}{n_2} \right).$$

**Conclusion.** With probability $1 - \delta$, we have $(\alpha, \varepsilon)$-distributional unlearning with

$$\alpha \geq \left( \frac{1}{2} - 3 \left( \frac{n_1 - f}{n_2} \right)^2 \right) \text{KL}(p_1 \parallel p_2) - \frac{3 \ln(4/\delta)}{2n_2} \left( 1 + \frac{n_1 - f}{n_2} \right),$$

$$\varepsilon \leq 3 \left( \frac{n_1 - f}{n_2} \right)^2 \text{KL}(p_1 \parallel p_2) + \frac{3 \ln(4/\delta)}{n_2} \left( 1 + \frac{n_1 - f}{n_2} \right).$$

$\square$

## A.4 SELECTIVE REMOVAL

**Lemma 6** (Dvoretzky–Kiefer–Wolfowitz Inequality)**.** *Let $x_1, x_2, \ldots, x_n$ be independent and identically distributed random variables with cumulative distribution function $F$. Define the empirical distribution function by*

$$\widehat{F}(t) = \frac{1}{n} \sum_{i=1}^{n} \mathbb{1}\{x_i \leq t\}.$$

*Then, for any $\delta \in (0, 1)$, with probability at least $1 - \delta$ we have*

$$\sup_{t \in \mathbb{R}} \left| \widehat{F}(t) - F(t) \right| \leq \sqrt{\frac{\ln(2/\delta)}{2n}}.$$

**Lemma 7.** *Let $\mu_1, \mu_2 \in \mathbb{R}$, $\sigma > 0$. Consider $n_1$ i.i.d. samples $x_1^{(1)}, \ldots, x_1^{(n_1)}$ from $\mathcal{N}(\mu_1, \sigma^2)$ and $n_2$ i.i.d. samples $x_2^{(1)}, \ldots, x_2^{(n_2)}$ from $\mathcal{N}(\mu_2, \sigma^2)$. We define $\hat{\mu}_2$ the average of the samples from $\mathcal{N}(\mu_2, \sigma^2)$, $\hat{\mu}_1$ the average of the $n_1 - f \leq n_1$ closest samples from $x_1^{(1)}, \ldots, x_1^{(n_1)}$ to $\hat{\mu}_2$. We define $F : t \in \mathbb{R} \mapsto \Phi(\frac{t - |\mu_1 - \mu_2|}{\sigma}) - \Phi(\frac{-t - |\mu_1 - \mu_2|}{\sigma})$, where $\Phi$ is the standard normal CDF.*

*For any $\delta \in (0, 1)$, we have with probability $1 - \delta$,*

$$|\hat{\mu}_1 - \mu_2| \leq F^{-1} \left( 1 - \frac{f}{n_1} + \sqrt{\frac{\ln(2/\delta)}{2n_1}} \right). \tag{14}$$

*Proof.* Recall from Equation (25) that $\hat{\mu}_1$ is the average of the $n_1 - f$ samples, out of $n_1$ i.i.d. from $p_1$, with the closest distance to $\hat{\mu}_2$, the empirical mean of $n_2$ samples from $p_2$.

Denote by $\hat{\tau}_f := |x_1^{(n_1-f:n_1)} - \hat{\mu}_2|$ the $(n_1 - f)$-th largest distance of $\hat{\mu}_2$ to $p_1$ samples. It is then immediate from the triangle inequality that

$$|\hat{\mu}_1 - \mu_2| = |\frac{1}{n_1 - f} \sum_{i=1}^{n_1-f} x_1^{(i:n_1)} - \mu_2| \leq \frac{1}{n_1 - f} \sum_{i=1}^{n_1-f} |x_1^{(i:n_1)} - \mu_2| \leq \hat{\tau}_f. \tag{15}$$

Besides, denoting by $\widehat{F}_1$ the empirical CDF of the empirical distribution over $\left\{ |x_1^{(i)} - \mu_2| : i \in [n_1] \right\}$, we have for all $t \in \mathbb{R}$:

$$\widehat{F}_1(t) = \frac{1}{n_1} \sum_{i=1}^{n_1} \mathbb{1}_{\left\{ |x_1^{(i)} - \mu_2| \leq t \right\}}. \tag{16}$$

Yet, we recall that with probability $1 - \frac{\delta}{2}$, we have

$$|\hat{\mu}_2 - \mu_2| \leq \sigma \sqrt{\frac{2\ln(4/\delta)}{n_2}}. \tag{17}$$

Therefore, the triangle inequality gives for $i \in [n_1]$, $|x_1^{(i)} - \mu_2| \leq |x_1^{(i)} - \hat{\mu}_2| + |\hat{\mu}_2 - \mu| \leq |x_1^{(i)} - \hat{\mu}_2| + \sigma \sqrt{\frac{2\ln(4/\delta)}{n_2}}$, and we deduce

$$\widehat{F}_1(t) = \frac{1}{n_1} \sum_{i=1}^{n_1} \mathbb{1}_{\left\{ |x_1^{(i)} - \mu_2| \leq t \right\}} \leq \frac{1}{n_1} \sum_{i=1}^{n_1} \mathbb{1}_{\left\{ |x_1^{(i)} - \hat{\mu}_2| \leq t - \sigma\sqrt{\frac{2\ln(4/\delta)}{n_2}} \right\}}. \tag{18}$$

In particular, by definition of $\hat{\tau}_f$, we have

$$\widehat{F}_1(\hat{\tau}_f + \sigma\sqrt{\frac{2\ln(4/\delta)}{n_2}}) \leq \frac{1}{n_1} \sum_{i=1}^{n_1} \mathbb{1}_{\left\{ |x_1^{(i)} - \hat{\mu}_2| \leq \hat{\tau}_f \right\}} = \frac{n_1 - f}{n_1} = 1 - \frac{f}{n_1}. \tag{19}$$

Now, observe that $|x_1^{(i)} - \mu_2|$ follows a folded normal distribution of location $\mu_1 - \mu_2$ and scale $\sigma^2$, since $x_1^{(i)}$ follows $p_1 = \mathcal{N}(\mu_1, \sigma^2)$. Denote by $F_1$ its CDF. Thanks to the Dvoretzky–Kiefer–Wolfowitz inequality (Lemma 6), we have with probability $1 - \frac{\delta}{2}$ that for all $t \in \mathbb{R}$,

$$|\widehat{F}_1(t) - F_1(t)| \leq \sqrt{\frac{\ln(4/\delta)}{2n_1}}. \tag{20}$$

Plugging the above in the previous inequality, and using a union bound, we get with probability $1 - \delta$,

$$F_1(\hat{\tau}_f + \sigma\sqrt{\frac{2\ln(4/\delta)}{n_2}}) \leq \widehat{F}_1(\hat{\tau}_f + \sigma\sqrt{\frac{2\ln(4/\delta)}{n_2}}) + \sqrt{\frac{\ln(4/\delta)}{2n_1}} \leq 1 - \frac{f}{n_1} + \sqrt{\frac{\ln(4/\delta)}{2n_1}}. \tag{21}$$

By taking the inverse $F_1^{-1}$ of the CDF $F_1$ and rearranging terms, we obtain with probability $1 - \delta$ that

$$\hat{\tau}_f \leq F_1^{-1}\left( 1 - \frac{f}{n_1} + \sqrt{\frac{\ln(4/\delta)}{2n_1}} \right) - \sigma\sqrt{\frac{2\ln(4/\delta)}{n_2}}. \tag{22}$$

Finally, going back to (15), we obtain with probability $1 - \delta$ that

$$|\hat{\mu}_1 - \mu_2| \leq \hat{\tau}_f \leq F_1^{-1}\left( 1 - \frac{f}{n_1} + \sqrt{\frac{\ln(4/\delta)}{2n_1}} \right) - \sigma\sqrt{\frac{2\ln(4/\delta)}{n_2}}. \tag{23}$$

$\square$

**Lemma 8.** *Let $\mu_1, \mu_2 \in \mathbb{R}$ and suppose that $x \sim \mathcal{N}(\mu_1, \sigma^2)$. Define the random variable $z := |x - \mu_2|$, with cumulative distribution function*

$$F_\sigma(t) := \mathbb{P}[z \le t] = \Phi\Big(\frac{t - |\mu_1 - \mu_2|}{\sigma}\Big) - \Phi\Big(\frac{-t - |\mu_1 - \mu_2|}{\sigma}\Big), \quad t \ge 0,$$

*where $\Phi$ denotes the standard normal CDF. Then, for any $p \in (0,1)$ the inverse CDF satisfies*

$$F_\sigma^{-1}(p) = \sigma \, g^{-1}\Big(p; \frac{|\mu_1 - \mu_2|^2}{2\sigma^2}\Big),$$

*where the function $g(u; \kappa)$ is defined by $g(u; \kappa) := \Phi(u - \sqrt{2\kappa}) + \Phi(u + \sqrt{2\kappa}) - 1$, and $g^{-1}(p; \kappa)$ denotes the inverse function in $u$ satisfying $g(u; \kappa) = p$. In particular, when $\mu_1 = \mu_2$ (so that $\kappa = 0$) we have $g(u; 0) = 2\Phi(u) - 1$ and thus $F_{0,1}^{-1}(p) = \Phi^{-1}\big(\frac{p+1}{2}\big)$.*

*Proof.* Since $x \sim \mathcal{N}(\mu_1, \sigma^2)$, we have that $z = |x - \mu_2|$ has CDF

$$F_\sigma(t) = \Phi\Big(\frac{t - |\mu_1 - \mu_2|}{\sigma}\Big) - \Phi\Big(\frac{-t - |\mu_1 - \mu_2|}{\sigma}\Big), \quad t \ge 0.$$

Introduce the change of variable $u = \frac{t}{\sigma}$ so that $t = \sigma \, u$. Then,

$$F_\sigma(\sigma \, u) = \Phi\Big(u - \frac{|\mu_1 - \mu_2|}{\sigma}\Big) - \Phi\Big(-u - \frac{|\mu_1 - \mu_2|}{\sigma}\Big).$$

Using the symmetry $\Phi(-x) = 1 - \Phi(x)$, this becomes

$$F_\sigma(\sigma \, u) = \Phi\Big(u - \frac{|\mu_1 - \mu_2|}{\sigma}\Big) + \Phi\Big(u + \frac{|\mu_1 - \mu_2|}{\sigma}\Big) - 1.$$

Defining $\kappa = \frac{|\mu_1 - \mu_2|^2}{2\sigma^2}$ and setting

$$g(u; \kappa) := \Phi(u - \sqrt{2\kappa}) + \Phi(u + \sqrt{2\kappa}) - 1,$$

we have $F_\sigma(\sigma \, u) = g(u; \kappa)$. Thus, if $u^*$ is the unique solution of $g(u^*; \kappa) = p$, then

$$F_\sigma(\sigma \, u^*) = p,$$

so that

$$F_\sigma^{-1}(p) = \sigma \, u^* = \sigma \, g^{-1}(p; \kappa).$$

In the special case $\mu_1 = \mu_2$ (so that $\kappa = 0$), we obtain $g(u; 0) = 2\Phi(u) - 1$, whose inverse is given by $u = \Phi^{-1}((p+1)/2)$. Hence, $F_1^{-1}(p) = \Phi^{-1}((p+1)/2)$, as required. $\square$

**Theorem 1** (Selective Removal). *Let $p_1, p_2 \in \mathcal{P}$ and $\delta \in (0,1)$. Let $f$ samples from $p_1$ be removed according to Selective Removal. With probability $1 - \delta$, the resulting estimate satisfies $(\alpha, \varepsilon)$-distributional unlearning with:*

$$\alpha \ge \frac{1}{2}\mathrm{KL}(p_1 \parallel p_2) - \frac{1}{2}\Big(\frac{n_1 - f}{n_2}\Big)^2 g^{-1}\Big(1 - \frac{f}{n_1} + \sqrt{\frac{\ln(4/\delta)}{2n_1}}; \mathrm{KL}(p_1 \parallel p_2)\Big)^2 - \frac{\ln(4/\delta)}{n_2},$$

$$\varepsilon \le \Big(\frac{n_1 - f}{n_2}\Big)^2 g^{-1}\Big(1 - \frac{f}{n_1} + \sqrt{\frac{\ln(4/\delta)}{2n_1}}; \mathrm{KL}(p_1 \parallel p_2)\Big)^2 + \frac{2\ln(4/\delta)}{n_2},$$

*where $g(u; \kappa) := \Phi(u - \sqrt{2\kappa}) + \Phi(u + \sqrt{2\kappa}) - 1$, for $u, \kappa > 0$, and $\Phi$ is the standard normal CDF.*

*Proof.* We recall that $p_1 = \mathcal{N}(\mu_1, \sigma^2), p_2 = \mathcal{N}(\mu_2, \sigma^2) \in \mathcal{P}$ and $p := \mathcal{N}(\mu, \sigma^2) \in \mathcal{P}$ are univariate Gaussian distributions. We are given $n_1$ i.i.d. samples $x_1^{(1)}, \ldots, x_1^{(n_1)}$ from $p_1$ and $n_2$ i.i.d. samples $x_2^{(1)}, \ldots, x_2^{(n_2)}$ from $p_2$.

The distance-based selection removes $f \le n_1$ selected samples from the target distribution $p_1$ with the $f$ largest distances to $\hat{\mu}_2 := \frac{1}{n_2}\sum_{i=1}^{n_2} x_2^{(i)}$ the empirical estimator of the mean of $p_2$. That is,

denoting by $x_1^{(1:n_1)}, \ldots, x_1^{(n_1:n_1)}$ the original $n_1$ samples from $p_1$ reordered by increasing distance to $\hat{\mu}_2$:

$$|x_1^{(1:n_1)} - \hat{\mu}_2| \leq \ldots \leq |x_1^{(n_1:n_1)} - \hat{\mu}_2|, \tag{24}$$

with ties broken arbitrarily, then distance-based selection retains only $x_1^{(1:n_1)}, \ldots, x_1^{(n_1-f:n1)}$ to obtain

$$\hat{\mu}_1 := \frac{1}{n_1-f} \sum_{i=1}^{n_1-f} x_1^{(i:n_1)}. \tag{25}$$

Subsequently, we set the center $\mu$ of the unlearned distribution $p$ to be:

$$\mu = \frac{(n_1-f)\hat{\mu}_1 + n_2\hat{\mu}_2}{n_1 - f + n_2}, \tag{26}$$

where $\hat{\mu}_1 = \frac{1}{n_1-f} \sum_{i=1}^{n_1-f} x_1^{(i:n_1)}$ and $\hat{\mu}_2 = \frac{1}{n_2} \sum_{i=1}^{n_2} x_2^{(i)}$. We also observe that a standard Hoeffding bound (Lemma 5) yields that:

$$|\hat{\mu}_2 - \mu_2| \leq \sigma \sqrt{\frac{2\ln(4/\delta)}{n_2}}, \tag{27}$$

with probability $1 - \frac{\delta}{2}$. We also recall that

$$\mathrm{KL}(p_1 \parallel p) = \frac{(\mu_1 - \mu)^2}{2\sigma^2}, \quad \mathrm{KL}(p_2 \parallel p) = \frac{(\mu_2 - \mu)^2}{2\sigma^2}. \tag{28}$$

**Preservation bound.** First, we upper bound the KL divergence of $p_2$ from $p$. To do so, we first use the triangle inequality to get

$$|\mu - \mu_2| = \left| \frac{(n_1-f)\hat{\mu}_1 + n_2\hat{\mu}_2}{n_1 - f + n_2} - \mu_2 \right| = \left| \frac{n_1-f}{n_1-f+n_2}(\hat{\mu}_1 - \mu_2) + \frac{n_2}{n_1-f+n_2}(\hat{\mu}_2 - \mu_2) \right|$$
$$\leq \frac{n_1-f}{n_1-f+n_2}|\hat{\mu}_1 - \mu_2| + \frac{n_2}{n_1-f+n_2}|\hat{\mu}_2 - \mu_2|.$$

Therefore, using (27) we have with probability $1 - \frac{\delta}{2}$:

$$|\mu - \mu_2| \leq \frac{n_1-f}{n_1-f+n_2}|\hat{\mu}_1 - \mu_2| + \frac{n_2}{n_1-f+n_2}\sigma\sqrt{\frac{2\ln(4/\delta)}{n_2}}.$$

Moreover, we know from Lemma 7 that with probabilty $1 - \frac{\delta}{2}$

$$|\hat{\mu}_1 - \mu_2| \leq F^{-1}\left( 1 - \frac{f}{n_1} + \sqrt{\frac{\ln(4/\delta)}{2n_1}} \right).$$

Using the above in the previous inequality with a union bound, yields that with probability $1 - \delta$

$$|\mu - \mu_2| \leq \frac{n_1-f}{n_1-f+n_2}F^{-1}\left( 1 - \frac{f}{n_1} + \sqrt{\frac{\ln(4/\delta)}{2n_1}} \right) + \frac{n_2}{n_1-f+n_2}\sigma\sqrt{\frac{2\ln(4/\delta)}{n_2}}.$$

We can further simplify the above using Lemma 8, which implies that for all $p > 0$

$$F^{-1}(p) = \sigma\, g^{-1}\left( p; \frac{|\mu_1 - \mu_2|^2}{2\sigma^2} \right),$$

where the function $g$ is defined by $g(u; \kappa) := \Phi(u - \sqrt{2\kappa}) + \Phi(u + \sqrt{2\kappa}) - 1$, for $u, \kappa > 0$. Plugging this in the previous bound yields

$$|\mu - \mu_2| \leq \frac{n_1-f}{n_1-f+n_2}\sigma\, g^{-1}\left( 1 - \frac{f}{n_1} + \sqrt{\frac{\ln(4/\delta)}{2n_1}}; \frac{|\mu_1 - \mu_2|^2}{2\sigma^2} \right) + \frac{n_2}{n_1-f+n_2}\sigma\sqrt{\frac{2\ln(4/\delta)}{n_2}}. \tag{29}$$

Dividing both sides by $\sigma\sqrt{2}$ and then using (28) yields with probability $1 - \delta$:

$$\sqrt{\mathrm{KL}(p_2 \parallel p)} \leq \frac{n_1 - f}{(n_1 - f + n_2)\sqrt{2}} g^{-1}\Big(1 - \frac{f}{n_1} + \sqrt{\frac{\ln(4/\delta)}{2n_1}}; \mathrm{KL}(p_1 \parallel p_2)\Big) + \frac{\sqrt{n_2 \log(4/\delta)}}{n_1 - f + n_2}. \tag{30}$$

The above directly implies, by taking squares and using Jensen's inequality and that $f \geq 0$, that with probability $1 - \delta$:

$$\mathrm{KL}(p_2 \parallel p) \leq \Big(\frac{n_1 - f}{n_2}\Big)^2 g^{-1}\Big(1 - \frac{f}{n_1} + \sqrt{\frac{\ln(4/\delta)}{2n_1}}; \mathrm{KL}(p_1 \parallel p_2)\Big)^2 + \frac{2\ln(4/\delta)}{n_2}. \tag{31}$$

**Removal bound.** Second, we lower bound the KL divergence of $p_1$ from $p$. To do so, we use Jensen's inequality and (29) to obtain that, with probability $1 - \delta$, we have

$$|\mu_1 - \mu_2|^2 = |\mu_1 - \mu + \mu - \mu_2|^2 \leq 2|\mu_1 - \mu|^2 + 2|\mu - \mu_2|^2$$

$$\leq 2|\mu_1 - \mu|^2 + 2\Big(\frac{n_1 - f}{n_1 - f + n_2}\Big)^2 \sigma^2 g^{-1}\Big(1 - \frac{f}{n_1} + \sqrt{\frac{\ln(4/\delta)}{2n_1}}; \frac{|\mu_1 - \mu_2|^2}{2\sigma^2}\Big)^2$$

$$+ \Big(\frac{n_2}{n_1 - f + n_2}\Big)^2 \sigma^2 \frac{4\ln(4/\delta)}{n_2}.$$

Rearranging terms, using that $f \geq 0$, and dividing by $4\sigma^2$ along with (28) yields that with probability $1 - \delta$ we have

$$\mathrm{KL}(p_1 \parallel p) = \frac{|\mu_1 - \mu|^2}{2\sigma^2} \geq \frac{|\mu_1 - \mu_2|^2}{4\sigma^2} - \frac{1}{2}\Big(\frac{n_1 - f}{n_2}\Big)^2 g^{-1}\Big(1 - \frac{f}{n_1} + \sqrt{\frac{\ln(4/\delta)}{2n_1}}; \frac{|\mu_1 - \mu_2|^2}{2\sigma^2}\Big)^2 - \frac{\ln(4/\delta)}{n_2}$$

$$= \frac{1}{2}\mathrm{KL}(p_1 \parallel p_2) - \frac{1}{2}\Big(\frac{n_1 - f}{n_2}\Big)^2 g^{-1}\Big(1 - \frac{f}{n_1} + \sqrt{\frac{\ln(4/\delta)}{2n_1}}; \frac{|\mu_1 - \mu_2|^2}{2\sigma^2}\Big)^2 - \frac{\ln(4/\delta)}{n_2}.$$

Now, using (28) we get

$$\mathrm{KL}(p_1 \parallel p) \geq \frac{1}{2}\mathrm{KL}(p_1 \parallel p_2) - \frac{1}{2}\Big(\frac{n_1 - f}{n_2}\Big)^2 g^{-1}\Big(1 - \frac{f}{n_1} + \sqrt{\frac{\ln(4/\delta)}{2n_1}}; \mathrm{KL}(p_1 \parallel p_2)\Big)^2 - \frac{\ln(4/\delta)}{n_2}$$

**Conclusion.** With probability $1 - \delta$, we have $(\alpha, \varepsilon)$-distributional unlearning with

$$\alpha \geq \frac{1}{2}\mathrm{KL}(p_1 \parallel p_2) - \frac{1}{2}\Big(\frac{n_1 - f}{n_2}\Big)^2 g^{-1}\Big(1 - \frac{f}{n_1} + \sqrt{\frac{\ln(4/\delta)}{2n_1}}; \mathrm{KL}(p_1 \parallel p_2)\Big)^2 - \frac{\ln(4/\delta)}{n_2},$$

$$\varepsilon \leq \Big(\frac{n_1 - f}{n_2}\Big)^2 g^{-1}\Big(1 - \frac{f}{n_1} + \sqrt{\frac{\ln(4/\delta)}{2n_1}}; \mathrm{KL}(p_1 \parallel p_2)\Big)^2 + \frac{2\ln(4/\delta)}{n_2}.$$

$\square$

## A.5 SIMPLIFIED SAMPLE COMPLEXITY FOR SELECTIVE REMOVAL

In this section, we can simplify the result of Theorem 1 on selective removal by simplifying cumbersome terms. This leads to Corollary 10, which then yields to the sample complexity results in Table 1.

We first prove an upper bound on the inverse CDF of a folded Normal for small quantiles.

**Lemma 9.** *Let* $\mu_1, \mu_2 \in \mathbb{R}, \sigma > 0$, *and* $x \sim \mathcal{N}(\mu_1, \sigma^2)$. *Define the random variable* $z = |x - \mu_2|$, *which follows a folded normal distribution whose cumulative distribution function (CDF) is given by*

$$F(t) := \mathbb{P}[z \leq t] = \Phi\Big(\frac{t - |\mu_1 - \mu_2|}{\sigma}\Big) - \Phi\Big(\frac{-t - |\mu_1 - \mu_2|}{\sigma}\Big), \quad t \geq 0, \tag{32}$$

*where $\Phi$ denotes the standard normal CDF. Then, for any $p > 0$ such that $F^{-1}(p) \leq |\mu_1 - \mu_2|$, it holds that:*

$$F^{-1}(p) \leq \frac{\sigma\, p}{\varphi\left(\frac{|\mu_1 - \mu_2|}{\sigma}\right)},$$

*where $\varphi(x) := \frac{1}{\sqrt{2\pi}} \exp\left(-\frac{x^2}{2}\right), x \in \mathbb{R}$, is the standard normal density.*

*Proof.* Since $F$, defined in (32), is continuously differentiable and strictly increasing on $[0, |\mu_1 - \mu_2|]$ (with $F(0) = 0$) and by the assumption $F^{-1}(p) \leq |\mu_1 - \mu_2|$, the Mean Value Theorem guarantees that there exists some $\xi \in [0, F^{-1}(p)]$ such that

$$F\left(F^{-1}(p)\right) = F(0) + F'(\xi)\left(F^{-1}(p) - 0\right).$$

Since $F(0) = 0$ and $F$ is strictly increasing, one may directly write, via the Mean Value Theorem, that there exists $\xi \in [0, F^{-1}(p)]$ with

$$F\left(F^{-1}(p)\right) = F'(\xi)\, F^{-1}(p).$$

By definition of the inverse CDF, $F\left(F^{-1}(p)\right) = p$; hence,

$$p = F'(\xi)\, F^{-1}(p).$$

It remains to lower-bound $F'(\xi)$ for $\xi \in [0, |\mu_1 - \mu_2|]$. We recall that for $t \geq 0$,

$$F(t) = \Phi\left(\frac{t - |\mu_1 - \mu_2|}{\sigma}\right) - \Phi\left(\frac{-t - |\mu_1 - \mu_2|}{\sigma}\right).$$

Taking the derivative with respect to $t$ yields

$$F'(t) = \frac{1}{\sigma}\, \varphi\left(\frac{t - |\mu_1 - \mu_2|}{\sigma}\right) + \frac{1}{\sigma}\, \varphi\left(\frac{-t - |\mu_1 - \mu_2|}{\sigma}\right)$$
$$= \frac{1}{\sigma}\, \varphi\left(\frac{t - |\mu_1 - \mu_2|}{\sigma}\right) + \frac{1}{\sigma}\, \varphi\left(\frac{t + |\mu_1 - \mu_2|}{\sigma}\right),$$

where we used that the standard normal density $\phi$ is symmetric.

For $t \in [0, |\mu_1 - \mu_2|]$, observe that $t - |\mu_1 - \mu_2| \leq 0$. Because the standard normal density is symmetric and nonincreasing on $[0, \infty)$, we have

$$\varphi\left(\frac{t - |\mu_1 - \mu_2|}{\sigma}\right) = \varphi\left(\frac{|\mu_1 - \mu_2| - t}{\sigma}\right) \geq \varphi\left(\frac{|\mu_1 - \mu_2|}{\sigma}\right).$$

Also, the second term $\varphi\left(\frac{t + |\mu_1 - \mu_2|}{\sigma}\right)$ is nonnegative. Hence, for all $t \in [0, |\mu_1 - \mu_2|]$ we have

$$F'(t) \geq \frac{1}{\sigma}\, \varphi\left(\frac{|\mu_1 - \mu_2|}{\sigma}\right).$$

In particular, at $t = \xi$ we obtain

$$F'(\xi) \geq \frac{1}{\sigma}\, \varphi\left(\frac{|\mu_1 - \mu_2|}{\sigma}\right).$$

Substituting this lower bound into the equation $p = F'(\xi)\, F^{-1}(p)$ yields

$$p \geq \frac{F^{-1}(p)}{\sigma}\, \varphi\left(\frac{|\mu_1 - \mu_2|}{\sigma}\right).$$

Rearranging the inequality gives the desired upper bound:

$$F^{-1}(p) \leq \frac{\sigma\, p}{\varphi\left(\frac{|\mu_1 - \mu_2|}{\sigma}\right)}.$$

This completes the proof. $\square$

We are now ready to prove the corollary below. Note that retaining dependences on $\alpha, \varepsilon$ only in this corollary leads to the result of Table 1.

**Corollary 10.** *Let $p_1, p_2 \in \mathcal{P}$ and $\delta \in (0,1)$. Let $f$ samples from $p_1$ be removed according to our distance-based scoring rule or at random before MLE. Then in each case, with probability at least $1 - \delta$, the resulting estimate satisfies $(\alpha, \varepsilon)$-distributional unlearning if:*

1. *Random removal:* $n_2 \geq \frac{12 \ln(4/\delta)}{\min\{\varepsilon, \alpha\}}$ *and* $\mathrm{KL}(p_1 \parallel p_2) \geq 8\alpha$, *and*

$$f \geq n_1 - n_2 \sqrt{\frac{2\mathrm{KL}(p_1 \parallel p_2) - \alpha}{12\mathrm{KL}(p_1 \parallel p_2)}}, \qquad f \geq n_1 - n_2 \min\left\{1, \sqrt{\frac{\varepsilon}{6\mathrm{KL}(p_1 \parallel p_2)}}\right\}.$$

2. *Selective removal:* $n_2 \geq 2\ln(4/\delta) \max\left\{\frac{1}{\varepsilon}, \frac{1}{\sqrt{\varepsilon}}, \frac{1}{\alpha}, \sqrt{\mathrm{KL}(p_1 \parallel p_2) - 4\alpha}\right\}$, $\mathrm{KL}(p_1 \parallel p_2) \geq 4\alpha$, *and* $f \geq n_1 \left(\frac{3}{2} + \sqrt{\frac{\ln(4/\delta)}{2n_1}} - \Phi(2\sqrt{2\mathrm{KL}(p_1 \parallel p_2)})\right)$, *and*

$$f \geq n_1 - \sqrt{n_1 n_2} \left(\frac{\varepsilon}{16\pi}\right)^{1/4} \exp(-\mathrm{KL}(p_1 \parallel p_2)),$$
$$f \geq n_1 - \sqrt{n_1 n_2} \left(\frac{\mathrm{KL}(p_1 \parallel p_2) - 4\alpha}{8\pi}\right)^{1/4} \exp(-\mathrm{KL}(p_1 \parallel p_2)).$$

*Proof.* We treat random and selective removal separately below.

**Random Removal.** Consider removing $f$ samples using the random removal mechanism, before maximum likelihood estimation. From Theorem 3, with probability $1 - \delta$, we achieve $(\alpha, \varepsilon)$-unlearning with:

$$\alpha \geq \left(\frac{1}{2} - 3\left(\frac{n_1 - f}{n_2}\right)^2\right) \mathrm{KL}(p_1 \parallel p_2) - \frac{3\ln(4/\delta)}{2n_2}\left(1 + \frac{n_1 - f}{n_2}\right) \qquad \text{(removal)},$$

$$\varepsilon \leq 3\left(\frac{n_1 - f}{n_2}\right)^2 \mathrm{KL}(p_1 \parallel p_2) + \frac{3\ln(4/\delta)}{n_2}\left(1 + \frac{n_1 - f}{n_2}\right) \qquad \text{(preservation)}.$$

Therefore, assuming that $n_2 \geq \frac{12\ln(4/\delta)}{\min\{\varepsilon, \alpha\}}$ and $\mathrm{KL}(p_1 \parallel p_2) \geq 8\alpha$, direct calculations show that it is sufficient to set:

$$f \geq n_1 - n_2 \sqrt{\frac{2\mathrm{KL}(p_1 \parallel p_2) - \alpha}{12\mathrm{KL}(p_1 \parallel p_2)}} \qquad \text{(removal)},$$

$$f \geq n_1 - n_2 \min\left\{1, \sqrt{\frac{\varepsilon}{6\mathrm{KL}(p_1 \parallel p_2)}}\right\} \qquad \text{(preservation)}.$$

**Selective Removal.** For selective removal, Theorem 1 shows that, with probability $1 - \delta$, we achieve $(\alpha, \varepsilon)$-unlearning with:

$$\alpha \geq \frac{1}{2}\mathrm{KL}(p_1 \parallel p_2) - \frac{1}{2}\left(\frac{n_1 - f}{n_2}\right)^2 g^{-1}\left(1 - \frac{f}{n_1} + \sqrt{\frac{\ln(4/\delta)}{2n_1}}; \mathrm{KL}(p_1 \parallel p_2)\right)^2 - \frac{\ln(4/\delta)}{n_2} \qquad \text{(removal)},$$

$$\varepsilon \leq \left(\frac{n_1 - f}{n_2}\right)^2 g^{-1}\left(1 - \frac{f}{n_1} + \sqrt{\frac{\ln(4/\delta)}{2n_1}}; \mathrm{KL}(p_1 \parallel p_2)\right)^2 + \frac{2\ln(4/\delta)}{n_2} \qquad \text{(preservation)}.$$

We can simplify these bounds using Lemma 9. Indeed, thanks to the latter and using the same notation and a simple change of variable, we have for all $p, \kappa > 0$ such that $p \leq F(|\mu_1 - \mu_2|)$

$$g^{-1}\left(p; \kappa\right) \leq \frac{p}{\phi(\sqrt{2\kappa})} = p\sqrt{2\pi}\exp(\kappa). \tag{33}$$

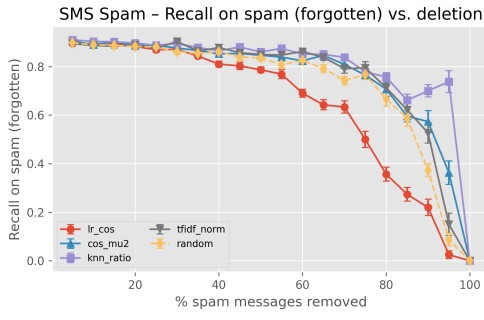
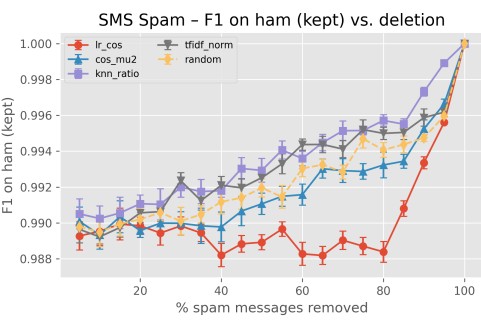

(a) Recall on spam ($p_1$, forgotten).        (b) Macro-*F1* on ham ($p_2$, kept).

Figure 5: **SMS Spam.** The likelihood-ratio score (LR-COS) pushes spam recall below $0.6$ after deleting $70\%$ of spam, whereas random deletion needs nearly $90\%$ removal to reach the same point. Ham performance remains almost flat ($<0.004$ absolute change) until the final $100$ % budget, affirming the tight preservation guarantee. Error bars: $\pm 1$ standard error over ten seeds.

Now, we plug in $p = 1 - \frac{f}{n_1} + \sqrt{\frac{\ln(4/\delta)}{2n_1}}$. Assuming $f \geq n_1\left(-\Phi(2\sqrt{2\mathrm{KL}(p_1 \parallel p_2)}) + \frac{3}{2} + \sqrt{\frac{\ln(4/\delta)}{2n_1}}\right)$, which directly implies that $p \leq F(|\mu_1 - \mu_2|)$ as required, we then obtain

$$g^{-1}\left(1 - \frac{f}{n_1} + \sqrt{\frac{\ln(4/\delta)}{2n_1}}; \mathrm{KL}(p_1 \parallel p_2)\right) \leq \left(1 - \frac{f}{n_1} + \sqrt{\frac{\ln(4/\delta)}{2n_1}}\right)\sqrt{2\pi}\exp\left(\mathrm{KL}(p_1 \parallel p_2)\right).$$

Plugging the above back in the first bounds due to Theorem 1, we obtain:

$$\alpha \geq \frac{1}{2}\mathrm{KL}(p_1 \parallel p_2) - \frac{1}{2}\left(\frac{n_1 - f}{n_2}\right)^2\left(1 - \frac{f}{n_1} + \sqrt{\frac{\ln(4/\delta)}{2n_1}}\right)^2 2\pi\exp\left(2\mathrm{KL}(p_1 \parallel p_2)\right) - \frac{\ln(4/\delta)}{n_2},$$

$$\varepsilon \leq \left(\frac{n_1 - f}{n_2}\right)^2\left(1 - \frac{f}{n_1} + \sqrt{\frac{\ln(4/\delta)}{2n_1}}\right)^2 2\pi\exp\left(2\mathrm{KL}(p_1 \parallel p_2)\right) + \frac{2\ln(4/\delta)}{n_2}.$$

Therefore, assuming that $n_2 \geq 2\ln(4/\delta)\max\left\{\frac{1}{\varepsilon}, \frac{1}{\sqrt{\varepsilon}}, \frac{1}{\alpha}, \sqrt{\mathrm{KL}(p_1 \parallel p_2) - 4\alpha}\right\}$ and $\mathrm{KL}(p_1 \parallel p_2) \geq 4\alpha$, and recalling we had assumed $f \geq n_1\left(-\Phi(2\sqrt{2\mathrm{KL}(p_1 \parallel p_2)}) + \frac{3}{2} + \sqrt{\frac{\ln(4/\delta)}{2n_1}}\right)$, direct calculations show that it is sufficient to set:

$$f \geq n_1 - \sqrt{n_1 n_2}\left(\frac{\varepsilon}{16\pi}\right)^{1/4}\exp(-\mathrm{KL}(p_1 \parallel p_2)) \qquad \text{(removal)},$$

$$f \geq n_1 - \sqrt{n_1 n_2}\left(\frac{\mathrm{KL}(p_1 \parallel p_2) - 4\alpha}{8\pi}\right)^{1/4}\exp(-\mathrm{KL}(p_1 \parallel p_2)) \qquad \text{(preservation)}.$$

$\square$

# B   EXPERIMENTAL DETAILS AND ADDITIONAL RESULTS

## B.1   SMS SPAM: A CONTENT-MODERATION UNLEARNING TASK

We revisit the UCI SMS Spam Collection (Almeida et al., 2011), treating the *spam* class ($p_1$) as information to forget and the *ham* class ($p_2$) as information to preserve. Messages are vectorised with TF–IDF features. Deletion budgets again span $5\%$ to $100\%$ of the spam slice in 5-point increments. We compare the same five scoring rules as before (COS-MU2, LR-COS, KNN-RATIO, TFIDF-NORM,

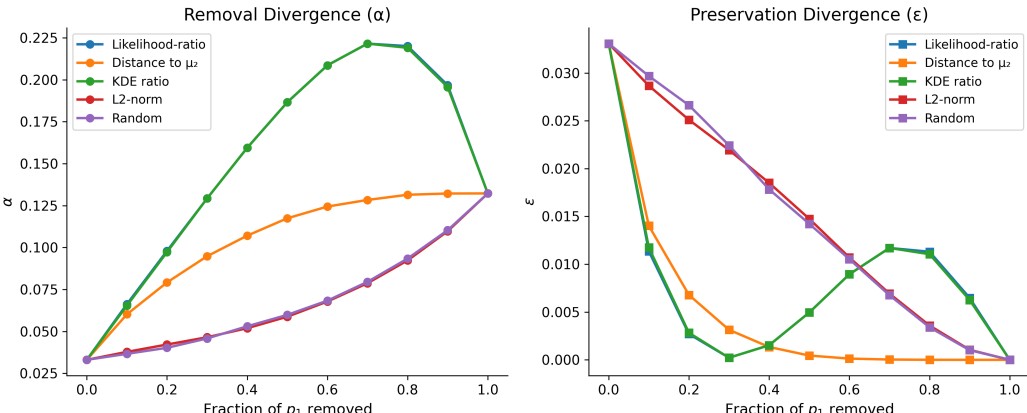

Figure 6: **Synthetic Gaussians.** Comparison of all strategies considered in the real-world datasets, using the low-divergence scenario of synthetic Gaussians (left plot, Fig. 2). Likelihood-ratio surpasses the distance-based strategy for lower deletion budgets. However, for larger budgets, it sacrifices utility for excessive removal. This is likely because it deletes samples representative of $p_1$ but close to $p_2$ since $p_1$ and $p_2$ are similar here.

RANDOM) and average results over ten random seeds. Metrics reported are *recall* on $p_1$ and macro-*F1* on $p_2$.

**Findings.** We report our main findings on our SMS Spam task in Figure 5. We observe that spam recall decays gradually until 75–80% deletion, after which all methods converge to zero as $p_1$ vanishes. LR-COS consistently dominates: it reaches a recall of $0.60$ at the 70% deletion budget, whereas random deletion does not cross that threshold until 90% deletion. Throughout, ham macro-$F_1$ *increases* slightly (see Fig. 5b), an artefact of class-imbalance—removing spam reduces false positives in the ham slice—yet the difference across methods never exceeds $0.002$. These results strengthen the evidence that selective deletion offers a $1.3-1.5\times$ sample-efficiency gain over random removal while preserving downstream utility almost perfectly.

**Remark 11** (Sample vs. Computational Efficiency in Unlearning). *The computational cost of sample-level unlearning typically scales with the size of the forget set. For instance, influence-function-based approaches, such as the one proposed by Guo et al. (2020), require computing gradients and Hessian-vector products for forget samples to approximate a second-order update. Other methods, like SalUn (Fan et al., 2024), identify and modify salient model parameters by backpropagating through the network for each forget sample. Reducing the size of the forget set implies less gradient computations and hence time savings. This forget size scaling observation also holds for the large class certified unlearning methods based on noise addition (Chourasia & Shah, 2023; Allouah et al., 2025; Waerebeke et al., 2025).*

## B.2 EXPERIMENTAL DETAILS

### B.2.1 HEURISTIC DELETION STRATEGIES

Across all datasets, we evaluate five scoring strategies for ranking samples in the forget distribution $p_1$ for removal. These scores approximate statistical dissimilarity from the retained distribution $p_2$ and correspond to different operational interpretations of divergence:

- **LR-COS / LR-MAHA** (likelihood-ratio inspired): A proxy for the log-likelihood ratio of $x$ under $p_2$ versus $p_1$:

$$s(x) = d(x, \mu_2) - d(x, \mu_1)$$

  where $d(\cdot, \mu)$ denotes cosine distance in TF–IDF space (text) or Mahalanobis distance in CNN feature space (images). Points are scored high when they are far from $p_2$ and close to $p_1$.

- **COS-MU2 / MAHA-MU2** (dissimilarity to $p_2$): Measures the distance from each $x \in p_1$ to the empirical mean of $p_2$:
$$s(x) = d(x, \mu_2)$$
  This approximates the contribution of each point to the KL divergence $\mathrm{KL}(p_2 \| \hat{p})$ when $p$ is modeled as a Gaussian.

- **KNN-RATIO** (local density ratio): Estimates the ratio of $k$-NN densities:
$$s(x) = \frac{\widehat{p_1}(x)}{\widehat{p_2}(x)}$$
  where $\widehat{p_i}(x) = \exp(-\|x - \mathrm{NN}_k^{(i)}(x)\|^2 / \sigma^2)$ is a local Gaussian kernel density using $k = 10$ nearest neighbors. This captures how typical $x$ is under $p_1$ versus $p_2$.

- **TFIDF-NORM / L2-NORM**: Uses the $\ell_2$ norm of the raw input (TF–IDF or real-valued) as a proxy for informativeness or deviation from the origin:
$$s(x) = \|x\|_2$$

- **CORESET** (centroid-based): Selects samples closest to the centroid of the forget set $p_1$:
$$s(x) = \|x - \mu_1\|_2,$$
  where $\mu_1$ is the empirical mean of $p_1$. This strategy prioritizes representative samples that are central to the forget distribution, potentially capturing the most characteristic patterns.

- **K-CENTER** (k-center greedy): Greedy selection to maximize minimum distance coverage of the forget set:
$$s(x) = \max_{i \in S} \|x - x_i\|_2,$$
  where $S$ is the set of already selected points. This strategy iteratively selects points that maximize the minimum distance to previously selected points, ensuring diverse coverage of the forget distribution.

- **RANDOM**: Samples points uniformly at random from $p_1$ as a baseline.

The first two sets of heuristics are direct extensions of our Gaussian selective removal method. When reporting results under "Selective removal" in tables, we report the best-performing heuristic among these.

**Remark 12** (Computational complexity of selection). *The selection methods evaluated present a spectrum of computational complexities, which is a key practical consideration for their use. At the most efficient end, methods like Random Removal, L2-Norm scoring, and the centroid-based Coreset are computationally inexpensive. Their complexity scales roughly linearly with the number of forget samples ($n_1$) and feature dimensions ($d$), making their cost approximately $O(n_1 d)$ plus sorting time. Our primary selective algorithms, based on distance to the retained set's mean, are moderately more complex. The Cosine distance variant remains efficient at $O((n_1 + n_2)d)$, but the Mahalanobis distance version is more demanding. Its need to compute an inverse covariance matrix introduces a term that scales with the cube of the dimensions ($O(d^3)$), making it potentially prohibitive for high-dimensional data.*

*On the other hand, some heuristics are computationally intensive. The KNN-RATIO method is particularly costly because it requires performing a k-nearest neighbor search for every sample in the forget set, which scales poorly with large datasets. Similarly, the iterative nature of the K-CENTER greedy algorithm makes it much slower than single-pass scoring approaches. This highlights a fundamental trade-off: while more complex methods like KNN-RATIO aim to capture nuanced distributional information, their computational overhead might be impractical. Simpler, faster methods like calculating the distance to the retained mean often provide a strong, practical balance between the effectiveness of the selection and its computational feasibility.*

### B.2.2    TABLE 5: SAMPLE-LEVEL UNLEARNING METHODS

We evaluate five sample-level unlearning strategies that represent different approaches to machine unlearning:

- **RETRAINING** (oracle baseline):

  - **Description:** Complete retraining from scratch on the retained data only. It trains a new model from random initialization using only samples from $p_2$. It provides the theoretical upper bound for sample-level unlearning performance, representing the gold standard but with highest computational cost.
  - **Hyperparameters:** Adam optimizer, learning rate $1 \times 10^{-3}$, 10 epochs, batch size 128.

- **FINE-TUNING** (naive baseline):

  - **Description:** Continues training the pre-trained model on retained data only. We initialize with pre-trained weights, then train for additional epochs on non-removed samples.
  - **Hyperparameters:** Adam optimizer, learning rate $5 \times 10^{-4}$, 2 epochs, batch size 128.

- **SALUN** (saliency-based unlearning (Fan et al., 2024)):

  - **Description:** This method resets parameters with highest saliency for forget samples, then finetunes on retained data: (1) Compute saliency scores for forget samples, (2) Reset top-$k\%$ most salient parameters to initial values, (3) Finetune on retained data. It leverages parameter importance to selectively reset the most forget-relevant parameters while preserving general knowledge.
  - **Hyperparameters:** Adam optimizer, learning rate $5 \times 10^{-4}$, 3 epochs, batch size 128, topk_percent=0.2.

- **NEGGRAD+** (stochastic gradient descent ascent (Kurmanji et al., 2024)):

  - **Description:** This method simultaneously maximizes loss on forget samples while minimizing loss on retained samples with loss $\beta \times \text{retain\_loss} + (1-\beta) \times \text{forget\_loss}$. This is essentially SGDA to balance retention of $p_2$ knowledge against forgetting of $p_1$ samples through opposing gradient directions.
  - **Hyperparameters:** SGD optimizer, learning rate $1 \times 10^{-3}$, momentum 0.99, weight decay 0.1, 3 epochs, $\beta = 0.9$, batch size 128.

- **INFLUENCE FUNC.** (influence function-based approximation):

  - **Description:** Approximates the effect of removing forget samples using influence functions, then applies parameter updates without retraining: (1) Compute influence scores for forget samples using Hessian-vector products, (2) Apply parameter updates based on influence estimates. This is computationally expensive for larger models, so we only use it for the Gaussian experiment.
  - **Hyperparameters:** Adam optimizer, learning rate $1 \times 10^{-4}$, 2 epochs, batch size 128, damping factor 0.01.

### B.2.3 SYNTHETIC GAUSSIANS

We draw $n_1 = n_2 = 1{,}000$ samples from $p_1 = \mathcal{N}(0, 1)$ and $p_2 = \mathcal{N}(\mu_2, 1)$ for $\mu_2 \in \{0.5, 2.5, 5.0\}$, with 20 seeds. After computing scores using each strategy, we remove the top-$f$ fraction of $p_1$ points, fit a Gaussian $\mathcal{N}(\hat{\mu}, 1)$ to the retained data, and compute:

$$\alpha = \text{KL}(p_1 \| \hat{p}), \qquad \varepsilon = \text{KL}(p_2 \| \hat{p})$$

These metrics match the forward-KL objectives of removal and preservation. No predictive model is trained; results reflect pure distributional divergence. For completeness, in Figure 6, we plot a comparison of all strategies considered in the real-world datasets, using the low-divergence scenario of synthetic Gaussians (left plot, Fig. 2). Likelihood-ratio surpasses the distance-based strategy for lower deletion budgets. However, for larger budgets, it sacrifices utility for excessive removal. This is likely because it deletes samples representative of $p_1$ but close to $p_2$ since $p_1$ and $p_2$ are similar here. This indicates that no removal strategy strictly dominates all others across all divergence (between $p_1$ and $p_2$) scenarios.

### B.2.4 JIGSAW TOXIC COMMENTS

We use the Jigsaw Toxic Comment Classification dataset, with 140K examples filtered to length 5–200 tokens. We define $p_1$ as all training comments containing any of the keywords: "f*ck", "s*it", "d*mn", "b*tch", "a*s", and $p_2$ as the remaining comments. For each of 5 random seeds, we:

1. Stratified-split $p_1$, $p_2$ into 70/30 train/val.

2. Compute TF–IDF embeddings (40K max features, 1–2 grams, sublinear TF, min_df=5).

3. Score and remove $f$ of $p_1$ training points using each heuristic.

4. Downsample $p_2$ to 5× the remaining $p_1$ size.

5. Train an $\ell_2$-regularized logistic regression on the edited data.

6. Evaluate *Recall@$p_1$* and *F1@$p_2$* on the validation sets.

### B.2.5 SMS SPAM COLLECTION

We use the UCI SMS Spam dataset (5574 examples, 13.4% spam). We apply:

1. TF–IDF vectorization (20K features, 1–2 grams, stopword removal).

2. Scoring of spam ($p_1$) messages using each heuristic.

3. Removal of top-$f$ fraction of spam for each strategy.

4. Retrain a logistic regression classifier.

5. Evaluate *Recall@spam* and *F1@ham* on a held-out 20% test split.

We run 10 seeds and report mean $\pm$ standard error.

### B.2.6 CIFAR-10 CLASS REMOVAL

We treat the "cat" class as $p_1$ and the other 9 classes as $p_2$. We use the standard CIFAR-10 split (50K train, 10K test), and proceed as follows:

1. Train a 3-block CNN (Conv–BN–ReLU ×2 + MaxPool, widths 32–64–128, global avg pool + linear head) for 10 epochs on the full training set.

2. Extract features for all training images using the penultimate layer.

3. Compute Mahalanobis distance scores for each cat image ($p_1$) using:

$$s_{\mathrm{maha}}(x) = \sqrt{(x-\mu)^\top \Sigma^{-1}(x-\mu)}$$

where $\mu$ and $\Sigma$ are estimated from $p_2$.

4. Delete the top-$f$ fraction of cat images under each scoring method.

5. Retrain the same CNN architecture on the edited training set.

6. Evaluate:

$$\mathrm{Accuracy}_{\mathrm{cat}}, \qquad \mathrm{Accuracy}_{\mathrm{non\text{-}cat}}$$

on the test set. Results are averaged over 30 random seeds.

### B.2.7 COMPUTING ENVIRONMENT

All experiments were run on a HPE DL380 Gen10 equipped with two Intel(R) Xeon(R) Platinum 8358P CPUs running at 2.60GHz, 128 GB of RAM, a 740 GB SSD, and two NVIDIA A10 GPUs. Training for vision experiments was implemented in PyTorch, while text-based experiments used Scikit-learn. All experiments were conducted using a single GPU.

### B.3    ABLATION: SENSITIVITY TO FEATURE REPRESENTATIONS

To assess how sensitive our selective deletion procedure is to the choice of feature representation, we conducted an ablation study on CIFAR–10 where we varied only the feature extractor and kept the selection algorithm and training setup fixed. We compare five representations: the default CNN features used in the main paper, ResNet–18 features (both pretrained and trained from scratch), and raw pixel features. For each setting, we measure forget-set accuracy at a deletion budget of 60% (lower is better forgetting). The results are shown in Table 6.

Two main observations emerge. First, across standard learned representations (CNN, ResNet–18 scratch, ResNet–18 pretrained), the forgetting performance is very similar, suggesting that the method is robust once the feature extractor provides a reasonably expressive embedding space. Second, using raw pixels significantly weakens forgetting, confirming that feature quality matters, but not in a way that makes the method brittle to the specific backbone.

| Feature Extractor | Forget Set Accuracy (%) |
|---|---|
| Random (baseline) | $40.5 \pm 2.5$ |
| CNN (default) | $21.6 \pm 1.9$ |
| ResNet–18 (trained from scratch) | $20.4 \pm 2.1$ |
| ResNet–18 (pretrained) | $27.3 \pm 1.8$ |
| Raw pixels | $31.8 \pm 2.0$ |

Table 6: Forget set accuracy (mean $\pm$ s.e.m.) at 60% deletion for different feature extractors on CIFAR–10. Lower is better forgetting.

### B.4    ADDITIONAL CORESET BASELINES ON CIFAR-10

To further investigate the behavior of retain-agnostic coreset methods in our unlearning setting, we compared our selective removal strategy against two strong data subset selection baselines, CRAIG, GradMatch, and CCS, on CIFAR–10. All methods operate at a fixed deletion budget of 60% from the forget class, and we report the resulting forget-set accuracy (lower is better forgetting). While CRAIG and GradMatch slightly, though not significantly, improve over random deletion, they still operate purely on the forget distribution and cannot exploit contrast with the retain distribution. As shown in Table 7, our distributional selection substantially outperforms these baselines.

| Selection Method | Forget Set Accuracy (%) |
|---|---|
| Random | $39.2 \pm 1.9$ |
| CRAIG (Mirzasoleiman et al., 2020) | $36.3 \pm 2.7$ |
| GradMatch (Killamsetty et al., 2021) | $35.5 \pm 4.2$ |
| CCS (Zheng et al., 2023) | $40.9 \pm 4.1$ |
| Selective (ours) | $24.5 \pm 1.4$ |

Table 7: Forget set accuracy (mean $\pm$ s.e.m.) on CIFAR–10 at a 60% deletion budget from the forget class. Lower values indicate stronger unlearning. Our selective removal method, which leverages the retain distribution, outperforms all strong coreset baselines (CRAIG, GradMatch, CCS) and random deletion.

