# OpenReview forum: "Distributional Machine Unlearning via Selective Data Removal"
_ICLR.cc/2026/Conference — ICLR 2026 Poster_

### Official Review · Reviewer_w9hG · 2025-10-28

**Soundness:** 3
**Presentation:** 3
**Contribution:** 3
**Rating:** 8
**Confidence:** 2

**Summary:**

This paper presents an intuitive method to select a subset of the forget data to remove/unlearn to balance model utility and unlearning performance.

**Strengths:**

The method is intuitive, novel, and well presented. The claims are theoretically and empirically backed up.

What I find especially relevant as a contribution is that this method can be used together with existing unlearning methods, improving them via a better composition of retain and forget sets.

**Weaknesses:**

The computational overhead for large datasets (e.g. web-scale data) is problematic. As this will require approximations to have a feasible retain set size, the performance of the method when the retain set is only available in an approximate manner (e.g. representative samples of an LLM training set) would be of interest as the results in table 2 already show a significant performance difference from Gaussian -> more realistic datasets (SMS, etc.).

**Questions:**

Q1 See question in weaknesses.

Q2 How should your method be used to pick the correct forget set (size of f samples) in practice when pairing it with an unlearning method (as in Table 5.)?

Q3 (Out of interest, feel free to ignore) Do you envision a version of your method that works in a setting where only the forget set is available?

---

> ### Author Response · Authors · 2025-11-19
> **Rebuttal**
>
> We thank the reviewer for the encouraging and insightful review. We have updated the submission to address their concerns.
>
> > The computational overhead for large datasets (e.g. web-scale data) is problematic. As this will require approximations to have a feasible retain set size, the performance of the method when the retain set is only available in an approximate manner (e.g. representative samples of an LLM training set) would be of interest as the results in table 2 already show a significant performance difference from Gaussian -> more realistic datasets (SMS, etc.).
>
> We thank the reviewer for bringing up this good point. We agree that principled selection requires computing similarities between forget and retain samples. In practice, one may only have a finite small sample from the retain distribution. Our selection scores are computed using finite samples already (e.g., training sets for images or comments), while evaluation is done on separate test sets. With only a small subset of retain data, the similarity estimates become noisier but still provide a useful signal, and our framework applies unchanged. Extending the framework to settings like LLMs, where one might only have access to proxy “general capability” datasets, is a natural next step and will involve a trade-off between computational overhead (to obtain features or scores) and selection quality. We emphasize this in blue in the third paragraph of Section 4.2 of the updated submission.
>
> > Q2 How should your method be used to pick the correct forget set (size of f samples) in practice when pairing it with an unlearning method (as in Table 5.)?
>
> This is a very good question. The deletion budget can be chosen in (at least) two complementary ways:
>
> 1. From computational constraints: given a sample-level unlearning method, one can translate its cost scaling into a maximum acceptable forget-set size and target this as the budget.
>
> 2. From selection scores: one can inspect the cumulative distribution of selection scores and retain a subset of samples that covers, for example, a fixed fraction (e.g., 80%) of the total “influence weight,” analogous to choosing a coverage threshold.
>
> In the paper, we explore the full budget curve to illustrate trade-offs, but we agree that a practical guideline for selecting a concrete operating point is useful. We emphasize this in blue in Remark 2 in Section 4.2 of the updated submission.
>
> > Q3 (Out of interest, feel free to ignore) Do you envision a version of your method that works in a setting where only the forget set is available?
>
> We thank the reviewer for bringing up this point. If no retain distribution (even a proxy) is available, the problem degenerates to selecting influential samples within a single distribution, which is closer to classical coreset construction. Our experiments in Section 4.3 suggest that such single-distribution approaches are limited in low-divergence settings: they tend to focus on representativeness within the forget distribution and cannot exploit the crucial contrast with the retain distribution.
>
> In practice, we hypothesize that even a coarse or proxy retain dataset (e.g., general capability data) helps instantiate the distributional unlearning objective as we define it. We emphasize this in blue in Remark 3 in Section 4.3 of the updated submission.

---

> > ### Comment · Reviewer_w9hG · 2025-11-25
> >
> > Thank you for the clarifications. As before, I am recommending acceptance (8).
> > No further questions from my side.

---

### Official Review · Reviewer_KqBQ · 2025-10-31

**Soundness:** 2
**Presentation:** 2
**Contribution:** 2
**Rating:** 2
**Confidence:** 3

**Summary:**

This paper introduces "distributional unlearning," a novel information-theoretic framework to address the problem of removing entire domains or subpopulations of data from a training set. The authors argue that full domain removal is computationally prohibitive, while random partial removal is statistically ineffective. Their key insight is that a domain's statistical influence is often concentrated in a small subset of its data. The proposed framework formalizes this by modeling the "forget" and "retain" data as two distinct probability distributions, p1 and p2. The goal is to find a data subset that maximally increases the KL divergence from p1 (removal) while minimally increasing the KL divergence from p2 (preservation). The paper provides strong theoretical results, deriving the removal-preservation Pareto frontier for exponential families and proving downstream log-loss guarantees for models trained on the resulting data. It proposes a simple and efficient distance-based algorithm—removing p1 samples that are furthest from the mean of p2—and proves it is quadratically more sample-efficient than random removal in low-divergence regimes. Experiments on synthetic, text, and image data validate the theory, showing that this selective approach can achieve strong unlearning effects with 15-82% fewer data deletions than full removal.

**Strengths:**

The paper's primary strength is its elegant formalization of domain unlearning as a distributional problem. By defining the objectives using KL divergence and characterizing the optimal trade-off via a Pareto frontier, the authors provide a rigorous, data-centric foundation for a problem that is often treated with ad-hoc heuristics. The connection established in Proposition 2 between the data-level KL objectives and the downstream model's expected log-loss is particularly powerful, as it provides a clear theoretical justification for why this data selection approach should work.

**Weaknesses:**

1. Novelty:
The core algorithm—removing points from one set based on their distance to the mean of another—is mechanically very simple. The novelty does not lie in a complex algorithmic contribution but rather in the application of this simple heuristic to the unlearning problem and, most importantly, the new theoretical framework that justifies it. While the framework is novel, the method itself could be seen as an application of standard outlier-detection or data-cleaning principles.
2. Experiments:
- Dependence on Feature Space: The effectiveness of the distance-based removal heuristics (e.g., MAHA-MU2, LR-COS) is entirely contingent on the quality of the embedding space. The paper uses standard features (e.g., from a pre-trained CNN), but provides no analysis of how the performance would change with a different, perhaps less-optimal, feature representation. If the feature space does not effectively separate the "statistically influential" samples, the distance metric may fail to identify them.
- Limited Baselines: The paper effectively demonstrates superiority over random removal and a simple coreset baseline. However, the coreset baseline (removing points closest to the p1 centroid) is shown to be ill-suited for this dual-distribution problem. A comparison with more sophisticated data selection methods, even if from different domains (e.g., active learning), could have provided a broader context for the performance gains.
- Theory-Practice Gap: The strong, closed-form sample complexity results are derived under the assumption of Gaussian distributions. While the authors acknowledge this and show that the qualitative insights generalize, the quantitative gap between the predicted 82% savings on synthetic data and the 15-50% savings on real-world data is substantial. This highlights that the complexity of real data distributions significantly tempers the theoretical gains.
- Practical Usage: the authors could consider more general/modern settings like LLM SFT.
3. Additional Things:
- The Oracle of p1 and p2: The entire framework operates on the strong assumption that the forget (p1) and retain (p2) sets are already known and perfectly separated. In practice, identifying every sample belonging to an unwanted domain (e.g., "toxic language") is an extremely challenging and often ambiguous task in itself. The paper solves the important second step of this process (which subset to remove) but does not address the first, which is a major practical caveat.
- Simplicity of "Statistical Influence": The method operationalizes "statistical influence" as distance from the retained data's mean. This is effective for unimodal, Gaussian-like distributions but may be less effective for multi-modal or complex distributions, where influence might be related to local density or boundary effects rather than just distance to a single central point.

**Questions:**

How sensitive are the results to the choice of the feature space? For the CIFAR-10 experiment, if you were to use features from a different model (e.g., a ViT instead of a CNN) or a model trained on a different dataset, would you expect the set of "most influential" cat images to remain largely the same?

---

> ### Author Response · Authors · 2025-11-19
> **Rebuttal (1/2)**
>
> We thank the reviewer for their detailed comments and acknowledge the concerns raised. We address each in turn next, along with new experiments and updates to the submission.
>
> > How sensitive are the results to the choice of the feature space? For the CIFAR-10 experiment, if you were to use features from a different model (e.g., a ViT instead of a CNN) or a model trained on a different dataset, would you expect the set of "most influential" cat images to remain largely the same?
>
> > Dependence on Feature Space: The effectiveness of the distance-based removal heuristics (e.g., MAHA-MU2, LR-COS) is entirely contingent on the quality of the embedding space. The paper uses standard features (e.g., from a pre-trained CNN), but provides no analysis of how the performance would change with a different, perhaps less-optimal, feature representation. If the feature space does not effectively separate the "statistically influential" samples, the distance metric may fail to identify them.
>
> We thank the reviewer for bringing up this point, which we directly address with *new experimental results* in Table 6 (Section B.3) in our updated submission. To evaluate how much our method actually depends on the choice of representation, we ran an additional ablation on CIFAR-10 where we varied only the feature extractor and kept the rest of the pipeline fixed. At a 60% deletion budget, the forget set accuracies (lower is better forgetting) are: Random 40.5 ± 2.5, CNN 21.6 ± 1.9, ResNet-18 20.4 ± 2.1, ResNet-18 (pretrained) 27.3 ± 1.8, and raw pixels 31.8 ± 2.0.
>
> We make two observations: (1) across standard learned representations (CNN, ResNet-18 scratch, ResNet-18 pretrained), the forgetting performance is very similar, indicating that our method is robust once features are reasonably expressive; and (2) using raw pixels yields noticeably weaker forgetting, which confirms that feature quality matters, but not in a way that makes the method brittle to the precise backbone.
>
> > Limited Baselines: The paper effectively demonstrates superiority over random removal and a simple coreset baseline. However, the coreset baseline (removing points closest to the p1 centroid) is shown to be ill-suited for this dual-distribution problem. A comparison with more sophisticated data selection methods, even if from different domains (e.g., active learning), could have provided a broader context for the performance gains.
>
> To address the reviewer’s point about stronger coreset baselines, we added in Table 7 (Section B.4) new experiments with CRAIG (Mirzasoleiman et al., 2020) and GradMatch (Killamsetty et al,. 2021) on CIFAR-10 at a fixed 60% deletion budget. Both methods improve upon random removal but still operate solely on the forget distribution. Consistent with our analysis of simpler coreset methods in Section 4.3, they underperform our distributional selection, which leverages contrast with the retain distribution. Specifically, the forget-set accuracies are: Selective: 24.5 ± 1.4; GradMatch: 35.5 ± 4.2; CRAIG: 36.3 ± 2.7; Random: 39.2 ± 1.9.
>
> These results support the structural limitation we discuss: even sophisticated single-distribution coresets cannot capture cross-distribution influence, whereas our selection method is designed precisely for this setting. We emphasize this in blue at the end of Section 4.3 of the updated submission.
>
> > Novelty: The core algorithm—removing points from one set based on their distance to the mean of another—is mechanically very simple. The novelty does not lie in a complex algorithmic contribution but rather in the application of this simple heuristic to the unlearning problem and, most importantly, the new theoretical framework that justifies it. While the framework is novel, the method itself could be seen as an application of standard outlier-detection or data-cleaning principles.
>
> We appreciate the opportunity to clarify this point: our selection rule's simplicity is not a limitation but a deliberate design choice with two advantages: (1) Provable guarantees: Theorem 1 provides formal $(\alpha,\epsilon)$-distributional unlearning bounds with strict sample complexity improvements over random removal; (2) Generalizability: as our experiments show, the principle transfers across modalities (images, text, synthetic) without task-specific tuning. To our knowledge, no prior work provides provable sample-complexity benefits for distributional unlearning subset selection.

---

> > ### Comment · Reviewer_KqBQ · 2025-11-26
> > **More baselines and results are required**
> >
> > I appreciate your new baseline results. However, they are still outdated in 2025. Please add coreset selection baseline results on TDDS[1], CCS[2], and UNSEEN[3], etc.
> >
> > [1] Zhang X, Du J, Li Y, et al. Spanning training progress: Temporal dual-depth scoring (tdds) for enhanced dataset pruning[C]//Proceedings of the IEEE/CVF Conference on Computer Vision and Pattern Recognition. 2024: 26223-26232.
> > [2] Zheng H, Liu R, Lai F, et al. Coverage-centric Coreset Selection for High Pruning Rates[C]//The Eleventh International Conference on Learning Representations.
> > [3] Xu F, Wang S, et al. Rethinking Dataset Pruning From A Generalization Perspective[C]// The Future of Machine Learning Data Practices and Repositories at ICLR 2025

---

> ### Author Response · Authors · 2025-11-19
> **Rebuttal (2/2)**
>
> > Theory-Practice Gap: The strong, closed-form sample complexity results are derived under the assumption of Gaussian distributions. While the authors acknowledge this and show that the qualitative insights generalize, the quantitative gap between the predicted 82% savings on synthetic data and the 15-50% savings on real-world data is substantial. This highlights that the complexity of real data distributions significantly tempers the theoretical gains.
>
> The Gaussian assumption in our analysis is stated explicitly and is used to obtain closed-form Pareto frontiers and finite-sample guarantees. We fully agree that complex real-world datasets may not satisfy Gaussianity. However, the purpose of our theoretical contributions is to (i) characterize the fundamental limits and trade-offs of distributional unlearning and (ii) guide algorithm design (e.g., selecting points far from the retain data).
> This is why we validate the same selection principle experimentally on text, image, and synthetic tasks, where we still observe consistent data savings (e.g., Synthetic: 50-82%, CIFAR-10: 50%, Jigsaw: 15%) despite the distributional mismatch. We highlight in blue in Section 5 of the updated submission that closing this gap with sharper non-Gaussian theory is a natural direction for future work.
>
> > Practical Usage: the authors could consider more general/modern settings like LLM SFT.
>
> We thank the reviewer for bringing up this point. We agree that unlearning in the context of large language models and SFT is an exciting application. In this paper, we deliberately focus on discriminative tasks to establish a clean and tractable foundation for distributional unlearning, both theoretically and empirically. Our framework itself is model-agnostic: the selection operates on data, and the downstream learner can be any model family (including generative models or LLMs). Concretely, one could use LLM embeddings (e.g., final layer activations) as the representation space; Use held-out general capability datasets (e.g., C4, OpenWebText) as retain distribution; Apply our distance-based criterion in embedding space. We emphasize this as an important future direction in blue at the end of Section 5 of the updated submission.
>
> > The Oracle of p1 and p2: The entire framework operates on the strong assumption that the forget (p1) and retain (p2) sets are already known and perfectly separated. In practice, identifying every sample belonging to an unwanted domain (e.g., "toxic language") is an extremely challenging and often ambiguous task in itself. The paper solves the important second step of this process (which subset to remove) but does not address the first, which is a major practical caveat.
>
> The reviewer raises a valid point about the assumption that samples from both distributions are available. As stated in the paper, our contribution is the selection step once such candidate forget samples have been identified. This step is **complementary** to (upstream) detection and (downstream) model-update methods, and is currently underexplored.
>
> In practice, candidate forget sets arise naturally, e.g.:
> * Source- or concept-based removal: metadata or tags identify content from specific sources or concepts.
> * Bias or domain-shift mitigation: classifiers or heuristics flag candidate problematic examples.
> * Privacy or user-driven erasure: deletion requests specify concrete data to remove.
>
> Given such candidates, our method determines a small high-impact subset needed to achieve the distributional unlearning objective. We clarify this complementary role in Section 2 of the updated submission.
>
> > Simplicity of "Statistical Influence": The method operationalizes "statistical influence" as distance from the retained data's mean. This is effective for unimodal, Gaussian-like distributions but may be less effective for multi-modal or complex distributions, where influence might be related to local density or boundary effects rather than just distance to a single central point.
>
> We appreciate the point that statistical influence may be more complex than captured by our current selection rule. Our main claim is that influence is often concentrated and that a simple theoretically grounded selection rule can already exploit this to yield significant data savings. We agree that more sophisticated selection algorithms could better capture influence in complex multimodal distributions, for example, by incorporating local density or cluster structure instead of a single global mean. We emphasize that our work is intended as a foundational step towards such extensions, as highlighted in blue in the second paragraph of Section 4.2 of the updated submission.

---

> > ### Comment · Reviewer_KqBQ · 2025-11-26
> > **Results on LLM/MLLM**
> >
> > Thank you for your response. That said, given the current state of the field in 2025, I would expect to see more extensive experimental results on supervised fine-tuning (SFT) with both LLMs and MLLMs. For instance, incorporating a representative modern multimodal model such as Qwen2.5-VL into the SFT evaluation would significantly strengthen the paper’s relevance.
> >
> > Moreover, the comparison against established coreset selection baselines—particularly in the context of multiple MLLMs and LLMs—should be expanded and discussed more thoroughly. Without these additions, the practical applicability of the proposed method to today’s AI industry remains limited.
> >
> > If time or computational resources constrain the inclusion of new experiments, I would strongly encourage the authors to at least cite and critically discuss relevant recent works in this space. Should these revisions be adequately addressed, I would be willing to reconsider my score accordingly.
> >
> >
> > [1] Wang S, Jin X, Wang Z, et al. Data whisperer: Efficient data selection for task-specific llm fine-tuning via few-shot in-context learning. ACL 2025
> > [2] Li M, Zhang Y, Li Z, et al. From quantity to quality: Boosting llm performance with self-guided data selection for instruction tuning[C]//Proceedings of the 2024 Conference of the North American Chapter of the Association for Computational Linguistics: Human Language Technologies (Volume 1: Long Papers). 2024: 7602-7635.
> > [3] Li Y, Hui B, Xia X, et al. One-shot learning as instruction data prospector for large language models[C]//Proceedings of the 62nd Annual Meeting of the Association for Computational Linguistics (Volume 1: Long Papers). 2024: 4586-4601.

---

> > > ### Author Response · Authors · 2025-11-26
> > >
> > > > I appreciate your new baseline results. However, they are still outdated in 2025. Please add coreset selection baseline results on TDDS[1], CCS[2], and UNSEEN[3], etc.
> > >
> > > We thank the reviewer for the follow-up. For coreset baselines, we have added all requested citations to the updated submission and clarified their relationship to our setting in Section 4.3. We also added a *new experiment* with one of the suggested methods in Section B.4, CCS (Coverage-centric Coreset Selection), and found that it performs comparably to random deletion on CIFAR-10 at the same 60% deletion budget (40.9 ± 4.1 vs. 39.2 ± 1.9).
> > > We further recall that we had already included new experiments with two popular gradient-based coreset approaches (CRAIG, GradMatch), which similarly fall far short of our distributional selection method. As emphasized in the paper (Sec. 4.3), while suggested coreset methods are effective single-distribution pruning techniques, they are not designed for our distinct distributional unlearning objective, which requires leveraging contrast between the forget and retain distributions.
> > >
> > > > If time or computational resources constrain the inclusion of new experiments, I would strongly encourage the authors to at least cite and critically discuss relevant recent works in this space. Should these revisions be adequately addressed, I would be willing to reconsider my score accordingly.
> > >
> > > For LLM SFT, we have added citations to the suggested recent works (Data Whisperer, self-guided selection, instruction-data prospectors) and clarified how our distributional framework complements future SFT unlearning pipelines in Section 5. We stress that our primary contribution is foundational: we introduce a formal definition of distributional unlearning with finite-sample guarantees and closed-form Pareto frontiers, and we validate the resulting selection rules in controlled discriminative settings where ground-truth distributions and metrics are precisely measurable. The framework itself is model-agnostic, since selection operates on data representations rather than model internals, and can in principle be instantiated with LLM or MLLM embeddings in SFT pipelines. Full-scale SFT benchmarks across multiple modern LLMs/MLLMs are not feasible within the rebuttal period as the reviewer acknowledged, so in this revision we focused on making these connections explicit and positioning LLM/MLLM validation as a natural and important next step rather than the primary scope of this paper.
> > >
> > >
> > >
> > > We hope these updates address the reviewer’s concerns and clarify the contribution of our work.

---

> > > > ### Comment · Reviewer_KqBQ · 2025-11-27
> > > >
> > > > Thanks for your further explanation and revision. Here is one more suggestion: please add more related work on LLM SFT in the final version.
> > > >
> > > > I would like to update my scores to 6. Good work.

---

> > > > > ### Author Response · Authors · 2025-12-01
> > > > >
> > > > > Thank you for the updated score. We have added the suggested LLM SFT related work to Section 5 of the revised version.

---

### Official Review · Reviewer_CfVa · 2025-10-31

**Soundness:** 3
**Presentation:** 4
**Contribution:** 3
**Rating:** 8
**Confidence:** 3

**Summary:**

The paper introduces the concept of distributional unlearning, a framework for studying the statistical properties of data removal in training datasets. The authors claim that may be convenitent to reduce not all the data of the forget set but a subset of them, the question they are trying to answer is what is the minimal subset of data points to remove (the forget set) in order to make the resulting data distribution far from the distribution of the forget set, while remaining close to the distribution of the retain set.
To formalize this idea, the authors define the notion of $(\alpha, \varepsilon)$- distributional unlearning for a distribution $p$ with respect to two other distributions $p_1$ and $p_2$. A distribution $p$ satisfies $(\alpha, \varepsilon)$-distributional unlearning if it is at least $\alpha$-far from $p_1$ and at most $\varepsilon$-close to $p_2$, where "far" and "close" are measured in terms of the KL divergence between $p_1 or $p_2 and $p$.
Most of the theoretical analysis in the paper focuses on Gaussian distributions, where the authors characterize the Pareto frontier of achievable $(\alpha, \varepsilon)$ values. They also prove a finite-sample guarantee for achieving $(\alpha, \varepsilon)$ distributional unlearning using two data deletion mechanisms: random removal, where a fixed number of samples are deleted uniformly at random; and relective removal, where data points are assigned scores proportional to their distance from the mean of the retain distribution, and high-scoring points are removed.
The authors compare the performance of these two algorithms on several datasets and also discuss the synergy of their approach with other unlearning methods.

**Strengths:**

- The paper is very well written, with clear exposition and excellent presentation. The theoretical claims are sound.
- The introduction of distributional machine unlearning represents a novel and interesting contribution, offering a perspective on how data deletion can be studied from a statistical standpoint.
- The inclusion of coreset-based methods as an additional baseline is appropriate and well-motivated.

**Weaknesses:**

- The theoretical results are derived for data following two gaussian distributions (strong assumption) for two simple removal strategies. While these allow for closed-form analysis, they are not particularly practical for real-world unlearning.
- The use of Kullback–Leibler (KL) divergence between data distributions as the central measure does not directly capture the notion of “forgetting” at the model level. KL divergence quantifies average differences between data distributions, not how much information a trained model retains about specific samples.
Therefore, KL-based analysis provides a bound on distributional robustness or generalization under deletion, rather than a guarantee of behavioral or algorithmic unlearning. In other words, a model may still encode information about deleted data even if the KL divergence between underlying distributions is small.
- It would be interesting to see
- The paper is interesting from the thoeretical point of view and for the introduction of this problem. But is not clear how useful can be in practice since typically retraining from scratch is something one may want to avoid.
- For the removal for instance of the class "cat" it is not shown how effective is this removal on samples that are

Minor:
- The setting considered in this paper is conceptually quite different from standard machine unlearning. The authors themselves acknowledge this distinction. Rather than proposing an algorithmic method to efficiently remove the influence of specific samples from a trained model, the work studies what happens statistically when data are deleted and the model is retrained from scratch.
Consequently, it is unclear whether the term unlearning is fully appropriate for this setting. The framework primarily measures how retraining after deletion affects model parameters and distributions, rather than addressing unlearning as a computational or algorithmic problem.
- The abstract highlights the derivation of the exact removal–preservation Pareto frontier for exponential families as a key contribution. However, this result appears only in the appendix and is not discussed or emphasized in the main text. Given its limited exposition and lack of integration into the core narrative, featuring it prominently in the abstract may be misleading or disproportionate relative to its role in the paper.

**Questions:**

- In practical machine learning settings, can the Kullback–Leibler (KL) divergence between two data distributions $p_1$ and $p_2$  remain small even when the samples removed from $p_1$  are rare but highly influential, such that the resulting model still retains significant information about those removed samples?
 I'm thinking for instance of removing a few minority-class examples can shift the decision boundary a lot or in linear regression about a few points can dominate parameter estimates, but contribute little to the overall KL.
- How is the synergy with other unlearning methods implemented in practice? For example, are the “selective samples” identified according to the procedure described in Section B.2.6 (so with the score only depending on their class and their embedding) and then used as the input (forget set) for the subsequent unlearning algorithm? (instead of performing retraining)
- See also weaknesses

---

> ### Author Response · Authors · 2025-11-19
> **Rebuttal (1/2)**
>
> We thank the reviewer for the detailed and positive evaluation. We have updated the submission to address their concerns.
>
> > The theoretical results are derived for data following two gaussian distributions (strong assumption) for two simple removal strategies. While these allow for closed-form analysis, they are not particularly practical for real-world unlearning.
>
> The Gaussian assumption in our analysis is stated explicitly and is used to obtain closed-form Pareto frontiers and finite-sample guarantees. We fully agree that complex real-world datasets may not satisfy Gaussianity. However, the purpose of our theoretical contributions is to (i) characterize the fundamental limits and trade-offs of distributional unlearning and (ii) guide algorithm design (e.g., selecting points far from the retain data).
> This is why we validate the same selection principle experimentally on text, image, and synthetic tasks, where we still observe consistent data savings (e.g., Synthetic: 50-82%, CIFAR-10: 50%, Jigsaw: 15%) despite the distributional mismatch. We highlight in blue in Section 5 of the updated submission that closing this gap with sharper non-Gaussian theory is a natural direction for future work.
>
> > The use of Kullback–Leibler (KL) divergence between data distributions as the central measure does not directly capture the notion of “forgetting” at the model level. KL divergence quantifies average differences between data distributions, not how much information a trained model retains about specific samples. Therefore, KL-based analysis provides a bound on distributional robustness or generalization under deletion, rather than a guarantee of behavioral or algorithmic unlearning. In other words, a model may still encode information about deleted data even if the KL divergence between underlying distributions is small.
>
> This is a valuable observation. We agree that our framework focuses on the data selection aspect of unlearning and we do not claim model-level privacy guarantees. Our goal in this paper is to isolate the data selection problem and remain agnostic to the specific sample-level unlearning algorithm that will be applied afterwards. Yet, the metrics we report (e.g., accuracy or recall on the forget set) match widely-used empirical measures of forgetting in the literature. We agree that combining our selection with formal model-level unlearning guarantees (e.g., membership inference-based metrics or certified unlearning) is an important next step. We state this explicitly in the penultimate sentence of Section 5 in the updated submission.
>
> > The paper is interesting from the thoeretical point of view and for the introduction of this problem. But is not clear how useful can be in practice since typically retraining from scratch is something one may want to avoid.
>
> We thank the reviewer for bringing up this point. We agree that the selection procedure should not be tailored only to retraining-from-scratch. This is precisely why in Section 4.3 we pair our selection methods with several sample-level unlearning algorithms (NegGrad+, SalUn, influence-function approximations, finetuning) and observe consistent gains in deletion efficiency across them. More broadly, our framework treats these methods as a downstream “black box”: once the forget set is selected, any sample-level unlearning method can consume it.
>
> > How is the synergy with other unlearning methods implemented in practice? For example, are the “selective samples” identified according to the procedure described in Section B.2.6 (so with the score only depending on their class and their embedding) and then used as the input (forget set) for the subsequent unlearning algorithm? (instead of performing retraining)
>
> Yes, our intended pipeline is: (1) Selection chooses a small high-impact subset from the identified unwanted samples, then (2) Sample-level unlearning uses this subset as input to update the model.

---

> ### Author Response · Authors · 2025-11-19
> **Rebuttal (2/2)**
>
> > In practical machine learning settings, can the Kullback–Leibler (KL) divergence between two data distributions $p_1$ and $p_2$ remain small even when the samples removed from $p_1$ are rare but highly influential, such that the resulting model still retains significant information about those removed samples? I'm thinking for instance of removing a few minority-class examples can shift the decision boundary a lot or in linear regression about a few points can dominate parameter estimates, but contribute little to the overall KL.
>
> This is a very good point. We agree that KL divergence has limitations and that different tasks may favor different statistical divergences. We chose KL because it (i) leads to tractable analysis in Gaussian/exponential families and (ii) directly controls expected log-loss, giving clean downstream guarantees (Proposition 2). That said, our framework is structurally compatible with other divergences, and one could design task-dependent divergences (e.g., reweighting regions of the space) to prioritize subpopulations that are more critical for a given task. We added a short discussion explicitly pointing out this extension after Definition 1 in the updated submission.
>
> > The setting considered in this paper is conceptually quite different from standard machine unlearning. The authors themselves acknowledge this distinction. Rather than proposing an algorithmic method to efficiently remove the influence of specific samples from a trained model, the work studies what happens statistically when data are deleted and the model is retrained from scratch. Consequently, it is unclear whether the term unlearning is fully appropriate for this setting. The framework primarily measures how retraining after deletion affects model parameters and distributions, rather than addressing unlearning as a computational or algorithmic problem.
>
> We use the term *distributional* unlearning to emphasize that our objective is to erase the influence of a subpopulation at the distribution level, irrespective of the particular computational method employed (retraining, influence-based updates, etc.). We distinguish this from sample-level unlearning, which assumes a fixed forget set and focuses on approximating retraining. Our selection step solves the problem of which subset of a domain to unlearn so that downstream sample-level methods can be applied more efficiently. We are open to alternative terminology and clarify this distinction in blue in the second paragraph of the introduction in Section 1 of the updated submission.
>
> > The abstract highlights the derivation of the exact removal–preservation Pareto frontier for exponential families as a key contribution. However, this result appears only in the appendix and is not discussed or emphasized in the main text. Given its limited exposition and lack of integration into the core narrative, featuring it prominently in the abstract may be misleading or disproportionate relative to its role in the paper.
>
> We appreciate the suggestion and agree with your remark. We adjusted the abstract in blue in the updated submission to mention explicitly that the closed-form Pareto frontier is derived for Gaussians, while the extension to broader exponential families and the empirical validations are described in the main text.

---

> > ### Comment · Reviewer_CfVa · 2025-11-28
> > **Answer to Rebuttal**
> >
> > Thank you for the detailed clarifications and updates. These adequately address the points I raised. I maintain my original recommendation for acceptance.

---

### Official Review · Reviewer_UHoi · 2025-11-02

**Soundness:** 3
**Presentation:** 3
**Contribution:** 3
**Rating:** 6
**Confidence:** 3

**Summary:**

This paper tackles the problem of machine unlearning, whcih traditionally focuses on removing entire domains or subpopulations of data as opposed to individual data points. The core of this paper is that a domain's statistical influence is often concentrated in small subsets of its samples. Thus, the authors propose distributional unlearning: a framework to select a small data subset by maximizing the statistical distance from the forget distribution while minimizing the distance to the retain distribution by using KL divergence. Experiments are conducted on synthetic, text, and image datasets.

**Strengths:**

- The paper does a good job of defining and motivating the proposed "distribution unlearning" framework.
- The framework is built with a strong theoretical foundation:
  - Using KL divergence to define forget and retain objectives is simple yet effective.
  - The proposed selective removal algorithm is quite intuitive and is derived directly from the theoretical analysis. The authors also analyze theoretically how the selective removal algorithm is a strategy to maximizing the KL divergnce of the forget set while minimizing the KL divergence of the retain set.
- Experimental evaluations, while not conducted on large-scale data, is well-structured and quite comprehensive.

**Weaknesses:**

- The framework is built on the assumption that the unwanted and retained data sets are already known. While the authors assert that this could be done via some upstream process (e.g., keyword filtering), there could be other scenarios where such techniques do not work.
- I'm not entirely sure if the core motivation, that a domain's influence is concentrated in a small subset holds in the experimental results. For example, in CIFAR10 the removal is only observed after 50% deletion and in Jigsaw it is only observed after 80% deletion.

**Questions:**

I would appreciate if the authors addressed the few concerns raised in weaknesses.

---

> ### Author Response · Authors · 2025-11-19
> **Rebuttal**
>
> We thank the reviewer for the thoughtful and constructive comments. We address each concern next, along with new updates to the submission.
>
> > I'm not entirely sure if the core motivation, that a domain's influence is concentrated in a small subset holds in the experimental results. For example, in CIFAR10 the removal is only observed after 50% deletion and in Jigsaw it is only observed after 80% deletion.
>
> The reviewer raises a good point about interpreting performance across domains. Our experiments were designed to illustrate the difficulty logic discussed in the paper: the achievable savings depend on how separable, structured, or intertwined the forget and retain distributions are. We recall that our experiments report:
>
> * Synthetic Gaussians (low separability, structured): This setting matches the analytical assumptions exactly. Influence is highly concentrated, resulting in large savings (50–82%).
>
> * CIFAR-10 (high separability): Although unlearning an entire class is more challenging, the forget distribution is coherent and well-separated, yielding ≈50% savings at matched forget accuracy.
>
> * Jigsaw (low separability, intertwined): Toxic comments form a heterogeneous and intertwined domain with diffuse influence. Savings are correspondingly smaller (≈15%) but still exceed random removal while preserving utility.
>
> We now make this difficulty logic explicit in Section 4 and in Table 2 of the revised submission.
>
> > The framework is built on the assumption that the unwanted and retained data sets are already known. While the authors assert that this could be done via some upstream process (e.g., keyword filtering), there could be other scenarios where such techniques do not work.
>
> The reviewer raises a valid point about the assumption that samples from both distributions are available. As stated in the paper, our contribution is the selection step once such candidate forget samples have been identified. This step is **complementary** to (upstream) detection and (downstream) model-update methods, and is currently underexplored.
>
> In practice, candidate forget sets arise naturally, e.g.:
> * Source- or concept-based removal: metadata or tags identify content from specific sources or concepts.
> * Bias or domain-shift mitigation: classifiers or heuristics flag candidate problematic examples.
> * Privacy or user-driven erasure: deletion requests specify concrete data to remove.
>
> Given such candidates, our method determines a small high-impact subset needed to achieve the distributional unlearning objective. We clarify this complementary role in Section 2 of the updated submission.

---

### Author Response · Authors · 2025-12-01
**Summary**

We would like to briefly summarize our contributions and the changes made during the discussion period. We thank the reviewers for their feedback and engagement.

Our paper studies the problem of removing the influence of an entire subpopulation rather than individual records. We formulate this as *distributional* unlearning, give a precise mathematical definition, and prove finite-sample guarantees linking selective data removal to changes in downstream performance. We also derive the optimal trade-off curve between forgetting one distribution and preserving another for Gaussian distributions and extend it to exponential families. Our experiments across multiple modalities (synthetic Gaussians, CIFAR-10, and Jigsaw/SMS) show that only a carefully chosen fraction of the forget set often suffices to match the effect of full removal while maintaining accuracy on the retain distribution.

During the rebuttal, we addressed the main points raised by the reviewers:
* **Feature-space ablations:** In response to Reviewer KqBQ, we added feature-space ablations (pixels, CNN, ResNet), confirming that our selection method behaves consistently across representations.
* **Stronger coreset baselines:** In response to Reviewer KqBQ, we added CRAIG, GradMatch, and CCS as additional coreset baselines. These new results support our initial observation that single-distribution pruning methods are not well suited to our two-distribution unlearning problem, as they underperform our selection rule at matched deletion budgets.
* **Difficulty logic in experiments:** In response to Reviewer UHoi, we clarified the difficulty progression across datasets and marked this explicitly in Table 2.
* **Practical guidance:** In response to Reviewer w9hG, we clarified how our method behaves when only a limited or approximate retain set is available, and added guidance on choosing a practical deletion budget.
* **Future work and extensions:** Following suggestions by Reviewers CfVa and KqBQ, we expanded Section 5 to include related work on LLM supervised finetuning and described how our framework could be integrated into future LLM unlearning pipelines, as well as additional discussion on distributional assumptions and model-level guarantees.

We appreciate the opportunity to make our contributions and scope clearer, and we believe our framework opens several useful directions for future work on machine unlearning.

---

### Meta-Review · Area_Chair_UbGg · 2026-01-05

**Summary:**

This paper proposes a distributional unlearning framework that uses KL-based constraints to identify a small subset of samples as the forget set. After the rebuttal and discussion, all reviewers lean toward acceptance. Notably, Reviewer KqBQ, who initially recommends rejection, raises concerns about the impact of feature space choices, requests additional experiments on stronger baselines, and questions the novelty and significance of the method. The reviewer later acknowledges that these concerns were adequately addressed in the rebuttal. The authors also clarify other points raised during review, including the dependence of the theoretical results on Gaussian assumptions and the computational cost of the proposed method. Overall, I do not see any substantive concerns remaining. I therefore recommend acceptance.

**Reviewer Concerns:**

Reviewer KqBQ and Reviewer w9hG actively participated in the discussion and explicitly acknowledged that their concerns were addressed.

The remaining two reviewers' comments also seem to be either resolved in the rebuttal or clearly recognized as limitations (e.g., reliance on Gaussian assumptions) and appropriately deferred to future work.

**Reviewer Scores:**

Reviewer KqBQ explicitly indicated that the score will be increased to 6.

Reviewer w9hG clearly stated that the positive score of 8 will be maintained.

The other two reviewers also seem likely to keep their favorable evaluations, as there is no apparent reason for them to lower their scores.

---

### Decision · Program_Chairs · 2026-01-26

Accept (Poster)